# PROMPT DIFFUSION ROBUSTIFIES ANY-MODALITY PROMPT LEARNING

## ABSTRACT

Foundation models enable prompt-based classifiers for zero-shot and few-shot learning. Nonetheless, the conventional method of employing fixed prompts suffers from distributional shifts that negatively impact generalizability to unseen samples. This paper introduces *prompt diffusion*, which uses a diffusion model to gradually refine the prompts to obtain a customized prompt for each sample. Specifically, we first optimize a collection of prompts to obtain over-fitted prompts per sample. Then, we propose a prompt diffusion model within the prompt space, enabling the training of a generative transition process from a random prompt to its overfitted prompt. As we cannot access the label of a test image during inference, our model gradually generates customized prompts solely from random prompts using our trained, prompt diffusion. Our prompt diffusion is generic, flexible, and modality-agnostic, making it a simple plug-and-play module seamlessly embedded into existing prompt learning methods for textual, visual, or multi-modal prompt learning. Our diffusion model uses a fast ODE-based sampling strategy to optimize test sample prompts in just five steps, offering a good trade-off between performance improvement and computational efficiency. For all prompt learning methods tested, adding prompt diffusion yields more robust results for base-to-new generalization, cross-dataset generalization, and domain generalization in classification tasks tested over 15 diverse datasets.

## 1 INTRODUCTION

Foundation models trained on a diverse set of image-text pairs that encapsulate a virtually limitless vocabulary of real-world concepts (Radford et al., 2021b; Jia et al., 2021; Li et al., 2022a), have demonstrated remarkable adaptability across various downstream tasks (Lin et al., 2014; Li et al., 2022b; 2023a; Zhang et al., 2022b; 2024). These models perform zero-shot image classification by filling in a predefined prompt template (*e.g.*, "`a photo of a [CLASS]`") with specific class names for the text encoder. Despite their effectiveness in generalizing to new tasks, performance can be affected by minor alterations in the wording of prompt templates, (Zhou et al., 2021). Rather than manually creating hand-made prompts, several new prompt learning techniques in natural language processing (Lester et al., 2021; Liu et al., 2021) and computer vision (Zhou et al., 2021; 2022a; Jia et al., 2022; Khattak et al., 2023a; Roy & Etemad, 2024; Li et al., 2024e) have been suggested, which focus on learning a set of soft prompts with the aid of a small amount of labeled data. However, training a model with such deterministic prompts often results in overfitting, causing the model to focus too much on the training data, which affects its ability to generalize. These methods usually fail when a considerable distribution shift between training and test data leads to suboptimal generalization performance. We propose generating a distribution of prompts for each sample, employing a probabilistic approach that effectively incorporates visual (domain) information in a manner capable of learning and adaptation.

We are inspired by diffusion models (Song et al., 2020; Zhou et al., 2024) that have emerged as a powerful generative technique with broad applicability for tasks as diverse as image generation (Ho et al., 2020), video processing (Ho et al., 2022), and text genera-tion (Gong et al., 2022). The core principle behind diffusion involves an iterative refine-ment of the data distributions, transitioning from a simple initial distribution to the desired target distribution. This iterative improvement process transforms the simple initial distribu-tion into a series of sub-transformations, making it a versatile tool suitable for various tasks.

To the best of our knowledge, we are the first to introduce diffusion models into prompt learning. Our process of generating prompts through a diffusion model is depicted in Figure 1. Our prompt diffusion involves gradually refining the prompts with a diffusion transformer, which leads to the development of custom prompts tailored to each sample, thereby enhancing the accuracy of predictions and robustifying their generalization across downstream tasks.

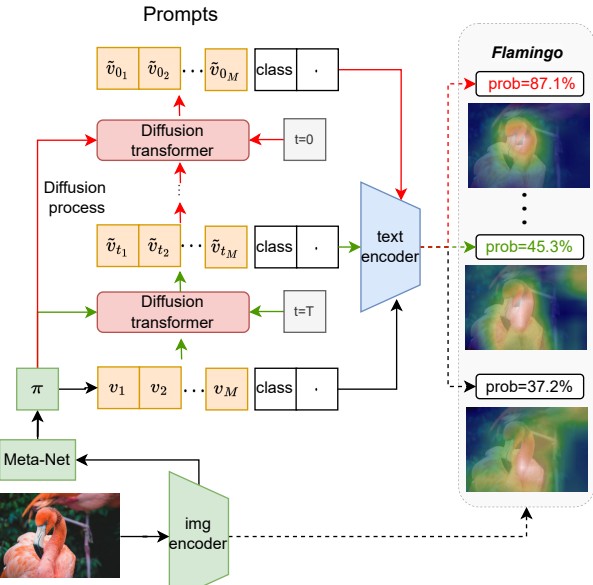

In this paper, we make three contributions. *First*, we propose a prompt diffusion method based on the transformer within the prompt space, enabling the learning of a generative pathway that seamlessly transitions from a random prompt to its personalized prompt. Rather than relying on a single static prompt acquired for the entire dataset, our prompt diffusion can learn and evolve from noise to the example prompt throughout the training process. These personalized prompts are adept at generalizing the unique domain characteristics inherent in each sample, thus enhancing

**Figure 1: Prompt diffusion** enhances traditional prompt learning methods such as CoCoOp (Zhou et al., 2022a) by introducing a diffusion process within the prompt space (colored arrows). Unlike deterministic prompt learning methods (black arrows), we employ a diffusion transformer to refine the prompts gradually. This process creates tailored prompts for each sample, complementing and augmenting existing prompting methods to achieve higher prediction accuracy through stronger generalization.

the model's ability to generalize. *Second*, to better deploy prompt diffusion, we propose a per-sample overfitting strategy to obtain "optima" prompts for each data sample, allowing our diffusion transformer to effectively navigate the transition from general to highly personalized prompts within the training phase. *Third*, our prompt diffusion approach is versatile, adaptable, and modality-agnostic, which makes it easily integrated as a plug-and-play module within existing prompt learning techniques. This includes methods specialized for text-based prompts (*e.g.*, CoCoOP (Zhou et al., 2022a)), visual prompts (*e.g.*, VPT (Jia et al., 2022)), as well as three approaches that combine both text and visual inputs (*e.g.*, MaPLe (Khattak et al., 2023a), PromptSRC (Khattak et al., 2023b), and CoPrompt (Roy & Etemad, 2024)). Our diffusion model leverages a state-of-the-art fast ODE-based sampling strategy (Zhou et al., 2024) that optimizes test sample prompts in just five steps, achieving an effective balance between performance enhancement and computational efficiency. To validate the effectiveness of our method, we conduct extensive testing across three common prompt learning experimental setups over 15 datasets: base-to-new generalization, cross-dataset generalization, and domain generalization. Adding prompt diffusion yields more robust results for all prompt learning methods tested.

## 2 RELATED WORK

**Foundation models.** Foundation models developed through training on a wide and varied collection of image-text pairs, capture a nearly boundless array of concepts from the real world (Radford et al., 2021b; Jia et al., 2021; Li et al., 2022a; Schneider et al., 2024; Xu et al., 2024), and have exhibited exceptional versatility in numerous downstream tasks (Lin et al., 2014; Li et al., 2022b; 2023a; Zhang et al., 2022b; 2024). Foundation models can be categorized into four types: 1) Masked language modeling, as investigated in studies such as (Kim et al., 2021; Lu et al., 2019), 2) Masked region prediction exemplified by (Tan & Bansal, 2019; Su et al., 2019), 3) Image-text matching addressed by works like (Tan & Bansal, 2019; Kim et al., 2021), and 4) Contrastive learning, with notable references including (Radford et al., 2021a; Jia et al., 2021; Li et al., 2021; Huo et al., 2021). Numerous studies have demonstrated improved performance in tasks such as few-shot image recognition (Gao et al., 2021; Zhang et al., 2022a; Kim et al., 2022), object detection (Li et al., 2024b; Maaz et al., 2022;

Zhou et al., 2022b; Gu et al., 2021; Zang et al., 2022; Cheng et al., 2024), and segmentation (Li et al., 2024d; Rao et al., 2022; Li et al., 2024c; Lüddecke & Ecker, 2022) using tailored methods. In this paper, we introduce a novel plugin designed to unify different prompt-learning approaches to address the issues of prompt engineering in traditional foundation models, aimed at solving the base-to-new, cross-dataset, and domain generalization of visual recognition problems.

**Prompt learning.** Prompt learning, originally introduced in the natural language processing community (Shin et al., 2020; Jiang et al., 2020; Liu et al., 2023a), involves applying a fixed function to input tokens to provide task instructions to the model. In the computer vision community, prompt learning has been explored in various forms, including textual prompts (Zhou et al., 2021; 2022a; Derakhshani et al., 2023; Lu et al., 2022b; Zhu et al., 2023; Liu et al., 2023b), visual prompts (Jia et al., 2022; Ge et al., 2022; Wang et al., 2022; Bahng et al., 2022; Li et al., 2024a; Yang et al., 2024), and multi-modal prompts (Khattak et al., 2023a; Lee et al., 2023; Li et al., 2023b; Roy & Etemad, 2024; Li et al., 2024e). 1) Textual prompt learning, as pioneered by CoOp (Zhou et al., 2021) and CoCoOp (Zhou et al., 2022a), fine-tunes a CLIP vision-language model (Radford et al., 2021a) for few-shot transfer by optimizing a continuous set of prompt vectors within its language branch. Bayesian prompt learning (Derakhshani et al., 2023) formulated prompt learning as a variational inference problem and demonstrated its ability to generalize unseen classes at the expense of base class accuracy. 2) Visual prompt tuning (Jia et al., 2022) introduces task-specific learnable prompts in the input visual space while keeping the pre-trained backbone fixed, optimizing them using the downstream task's label. 3) Multi-modal prompt learning (Khattak et al., 2023a;b; Li et al., 2024e; Xiao et al., 2024) applied prompt learning in both vision and language branches to improve the alignment between the vision and language representations. In contrast to previous prompt learning methods, this paper introduces modality-agnostic prompt diffusion, which leverages a diffusion model to generate prompts gradually. Our method serves as a simple plug-and-play module that seamlessly integrates with existing prompt learning methods, whether textual, visual, or multi-modal.

## 3 PRELIMINARIES

Before detailing our prompt diffusion, we first present the technical background on the CLIP foundation model, prompt-based learning, and diffusion models.

**Contrastive Language-Image Pre-Training (CLIP).** The objective of CLIP (Radford et al., 2021a) is to train an image encoder $f_I$ and a text encoder $g_T$ through contrastive pre-training using a large set of paired images and texts. This encourages the encoders to align corresponding image-text pairs in a shared semantic space. After pre-training, CLIP exhibits the capacity for zero-shot visual recognition by casting classification as an image-text matching task. Specifically, the term "`[CLASS]`" is utilized as a placeholder within a prompt template (*e.g.*, "`a photo of a [CLASS]`") for the text encoder $g_T$. If we let $g_T(\mathbf{T}_i)$ represent text features extended for class $i$, the classification probability for class $i$ given an image $\mathbf{I}$ is:

$$p(y{=}i|\mathbf{I}){=}\frac{\exp(\langle g_T(\mathbf{T}_i), f_I(\mathbf{I})\rangle/\tau)}{\sum_{j=1}^{K}\exp(\langle g_T(\mathbf{T}_j), f_I(\mathbf{I})\rangle/\tau)}, \tag{1}$$

where $\langle g_T(\mathbf{T}_i), f_I(\mathbf{I})\rangle$ denotes the cosine similarity between the image feature $f_I(\mathbf{I})$ and the class-specific text feature $g_T(\mathbf{T}_i)$ for the $i$-th class, $K$ the total number of classes, and $\tau$ the temperature parameter optimized during training.

**Prompt-based learning** enhances the transferability of the CLIP model by avoiding the need for prompt engineering. Instead, it enables automatic learning of prompts with a few samples from a downstream task. CoOp (Zhou et al., 2021) introduces and refines a set of $M$ continuous context vectors $\boldsymbol{V}{=}\{\boldsymbol{v}_1, \boldsymbol{v}_2, \ldots, \boldsymbol{v}_M\}$ as the learnable prompt. The prompt $\boldsymbol{T}_i{=}\{\boldsymbol{v}_1, \boldsymbol{v}_2, \ldots, \boldsymbol{v}_M, \boldsymbol{c}_i\}$ is a concatenation of the learnable context vectors $\boldsymbol{V}$ and the class token embedding $\boldsymbol{c}_i$, which is then inputted to the text encoder $g_T(\cdot)$. CoOp tailors the static context vectors $\boldsymbol{V}$ by minimizing the negative log-likelihood for the correct class token:

$$\mathcal{L}_{\text{CE}}(\boldsymbol{V}){=}-\sum_i \boldsymbol{y}_i \log p(\boldsymbol{T}_i|\boldsymbol{I}), \tag{2}$$

Here, $\boldsymbol{y}_i$ denotes the one-hot ground-truth label for class $i$. In downstream tasks, the pre-trained model parameters remain frozen, allowing the learnable prompt vectors $\boldsymbol{V}$ to be efficiently optimized through the minimization of the cross-entropy loss with only a limited number of samples.

**Diffusion model.** In denoising diffusion probabilistic models (Ho et al., 2020), $q(\mathbf{x}_t|\mathbf{x}_{t-1})$, is characterized as a Markov chain that progressively introduces Gaussian noise at each time step $t$, beginning with a clean image $\mathbf{x}_0 \sim q(\mathbf{x}_0)$. The *forward* diffusion process is formulated as:

$$q(\mathbf{x}_T|\mathbf{x}_0) := \prod_{t=1}^{T} q(\mathbf{x}_t|\mathbf{x}_{t-1}), \tag{3}$$

where $q(\mathbf{x}_t|\mathbf{x}_{t-1}) := \mathcal{N}(\mathbf{x}_t; \sqrt{1-\beta_t}\mathbf{x}_{t-1}, \beta_t\boldsymbol{I})$, $\{\beta\}_{t=0}^{T}$ is a variance schedule. By defining $\alpha_t := 1-\beta_t$ and $\bar{\alpha}_t := \prod_{s=1}^{t} \alpha_s$, the forward diffused sample at time step $t$, denoted as $\boldsymbol{x}_t$, can be generated in a single step as $\mathbf{x}_t = \sqrt{\bar{\alpha}_t}\mathbf{x}_0 + \sqrt{1-\bar{\alpha}_t}\boldsymbol{\epsilon}$.

The *reverse* process of the diffusion model learns to maximize the variational lower bound using parameterized Gaussian transitions, $p_\theta(\mathbf{x}_{t-1}|\mathbf{x}_t)$. Consequently, the reverse process is approximated as a Markov chain with the learned mean and fixed variance, starting from random noise $\mathbf{x}_T \sim \mathcal{N}(\mathbf{x}_T; \mathbf{0}, \boldsymbol{I})$. The diffusion model is trained by optimizing the following objective function:

$$\mathcal{L}_\theta = \mathbb{E}_{t,\mathbf{x}_0,\boldsymbol{\epsilon}}\left[\|\boldsymbol{\epsilon} - \boldsymbol{\epsilon}_\theta(\sqrt{\bar{\alpha}_t}\mathbf{x}_0 + \sqrt{1-\bar{\alpha}_t}\boldsymbol{\epsilon}, t)\|^2\right]. \tag{4}$$

In the *sampling* phase of diffusion, to sample from $p_\theta(\mathbf{x}_{t-1}|\mathbf{x}_t)$, one can perform the following:

$$\mathbf{x}_{t-1} = \frac{1}{\sqrt{\alpha_t}}\left(\mathbf{x}_t - \frac{1-\alpha_t}{\sqrt{1-\bar{\alpha}_t}}\boldsymbol{\epsilon}_\theta(\mathbf{x}_t, t)\right) + \sigma_t\boldsymbol{\epsilon}. \tag{5}$$

Based on the geometric property that each sampling trajectory approximately resides within a two-dimensional subspace embedded in a high-dimensional space, Zhou et al. (2024) introduce the Approximate MEan-Direction Solver (AMED-Solver), a single-step ODE solver that predicts the mean direction at each sampling step. By appropriately selecting $s_n$ and $c_n$, the AMED-Solver achieves an approximation given by:

$$\mathbf{x}_{t_n} \approx \mathbf{x}_{t_{n+1}} + c_n(t_n - t_{n+1})\epsilon_\theta(\mathbf{x}_{s_n}, s_n). \tag{6}$$

This formulation provides a single-step ODE solver, and the DPM-Solver-2 (Lu et al., 2022a) can be derived by setting $s_n = \sqrt{t_n t_{n+1}}$ and $c_n = 1$. Unlike typical approaches that operate on images, our prompt diffusion model directly optimizes prompts. Given that prompt learning in vision-language tasks aims for faster and more accurate image classification, our proposed prompt diffusion, built upon the AMED-Solver, enables more rapid image classification during inference time.

# 4 METHODOLOGY

This section outlines our approach to training prompt learning via our proposed prompt diffusion model. Our prompt diffusion model is an end-to-end framework that integrates the generation of sample-specific overfitted prompts with the diffusion process for prompt refinement. We begin by explaining how to generate sample-specific overfitted prompts in Section 4.1. Next, we introduce prompt diffusion during both the training and testing phases to obtain diffused prompts in Section 4.2.

## 4.1 PER-SAMPLE PROMPT OVERFITTING

Our approach begins by fine-tuning various prompts to achieve individualized overfitting for each data sample. This ensures the precise generation of prompts that are tailored to specific instances. Specifically, when dealing with an image represented as $\boldsymbol{x}$, we aim to obtain a set of prompts, denoted as $\boldsymbol{V}^*$, which have been explicitly overfitted to that sample. We feed both the image $\boldsymbol{x}$ and the initial prompts $\boldsymbol{V} = \{v_1, v_2, \ldots, v_M\}$ into various prompt learning models and then employ iterative gradient descent on Eq. (2) to optimize the set of prompts, resulting in $\boldsymbol{V}^* = \{v_1^*, v_2^*, \ldots, v_M^*\}$. These optimized prompts can be considered as the "optima" prompts for each sample. Note that the intermediate loss is solely adjusted to achieve overfitted prompts in this process. Afterward, the gradient information for the learnable prompts will be discarded without optimization incorporated into the final loss. We illustrate this per-sample prompt overfitting for textual prompt learning with CoCoOp (Zhou et al., 2022a) in Figure 2.

Once we obtain the overfitted prompts, our objective is to train the model using random prompts about these overfitted prompts. This is necessary because we cannot access the overfitted prompts

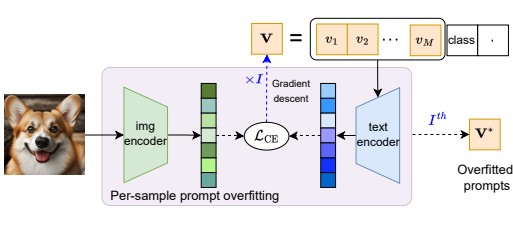

**Figure 2: Per-sample prompt overfitting** for textual prompt learning. Through a minimal number of iterations $I$ using gradient descent, we successfully derive overfitted prompts for each sample in the dataset. These overfitted prompts act as a "ground truth" for the prompts of each sample, enabling our proposed diffusion transformer to grasp the transition from generic prompts to highly personalized overfitted prompts.

during the testing stage. Therefore, in the following section, we use the diffusion model to learn the generative process of sample-specific prompts, thus robustifying the generalizability of the prompts for each sample.

## 4.2 PROMPT DIFFUSION

We leverage the diffusion model (Song et al., 2020) to model textual, visual, or multi-modal prompts. In our implementation, we adapt the diffusion process to incrementally denoise and refine overfitted prompts, thereby enhancing the generative quality and coherence of the prompts. Consequently, we introduce the concept of modality-agnostic prompt diffusion, a novel method that incrementally crafts sample-specific prompts for each instance. This methodical generation of prompts enhances their overall quality, ensuring that each prompt is optimally tuned to the nuances of its corresponding sample. This adaptive approach is designed to fine-tune the diffusion process, allowing for a more targeted and effective prompt generation that elevates the efficacy of the model's performance.

**Training phase.** During the initial training stage, we obtain the overfitted prompts $\boldsymbol{V}^*$ of individual samples via our proposed per-sample prompt overfitting. Then, the diffusion model is used to progressively approximate the overfitted prompts, from a Gaussian noise vector $\tilde{\boldsymbol{V}}_T \sim \mathcal{N}(0, \boldsymbol{I})$, which possesses the exact dimensions as $\boldsymbol{V}^*$. The approximation process iterates through the noise vectors $\tilde{\boldsymbol{V}}_t^*$, with $t$ representing the diffusion step from $T$ to 0. This process leads to the reconstruction of $\tilde{\boldsymbol{V}}_0$, which is expected to closely mirror the overfitted prompt associated with the particular sample being analyzed.

Specifically, throughout the forward diffusion phase at an increment in time $t$, we derive the overfitted prompts $\boldsymbol{V}_t^*$. Subsequently, the noised prompts, denoted as $\tilde{\boldsymbol{V}}_t$, and the training image feature $\pi$ - extracted through a lightweight neural network, Meta-Net $\pi(\theta)$ (Zhou et al., 2022a) - are utilized to create a conditional token for each input and the temporal timestep $t$. These are then inputted into the diffusion transformer. This process yields the interim diffused prompts $\tilde{\boldsymbol{V}}_t$. These prompts then, the token [CLASS] is synergized and integrated into the text encoder to generate the corresponding text features. The prediction of the final classification outcome for the training image is then conducted by Eq. (1). For each sample, our diffusion model encapsulates a dual-component objective comprising the variational lower bound $\mathcal{L}_{\text{diff}}$ for the diffusion model and the cross-entropy loss $\mathcal{L}_{\text{CE}}$. The overarching schema of our training scheme is depicted at the top of Figure 3.

The objective function, the simplified variational lower bound, aims to predict the denoised overfitted prompts accurately. Formally, the loss function is given by:

$$\mathcal{L}_{\text{diff}} = \left\| \boldsymbol{V}^* - \tilde{\boldsymbol{V}}_\theta \left( \sqrt{\bar{\alpha}_t} \boldsymbol{V}^* + \sqrt{1 - \bar{\alpha}_t} \boldsymbol{\epsilon}, \pi, t \right) \right\|^2, \tag{7}$$

where $\tilde{\boldsymbol{V}}_\theta(\cdot, \cdot, \cdot)$ denotes the function parameterized by the transformer architecture (Vaswani et al., 2017). This function processes the input comprising the original overfitted prompts $\boldsymbol{V}^*$, image feature $\pi$, and the diffusion time step $t$. The efficacy of our model is measured by its ability to minimize this loss, thus accurately reconstructing the overfitted prompts from their noised counterparts. By utilizing Eq. (2), we derive the final prediction $\hat{\boldsymbol{y}}$ using diffused prompts $\tilde{\boldsymbol{V}}_t$. The final objective is:

$$\mathcal{L}_{\text{final}} = \sum_{(x,y)} \Big[ - \mathbb{E}_{q(\tilde{\boldsymbol{V}}_t | \tilde{\boldsymbol{V}}_{t+1}, \pi)} \big[ \log p(\mathbf{y} | \mathbf{x}, \tilde{\boldsymbol{V}}_t) \big]$$
$$+ \beta \left\| \boldsymbol{V}^* - \tilde{\boldsymbol{V}}_\theta \left( \sqrt{\bar{\alpha}_t} \boldsymbol{V}^* + \sqrt{1 - \bar{\alpha}_t} \boldsymbol{\epsilon}, \pi, t \right) \right\|^2, \tag{8}$$

where $\beta$ represents a hyperparameter.

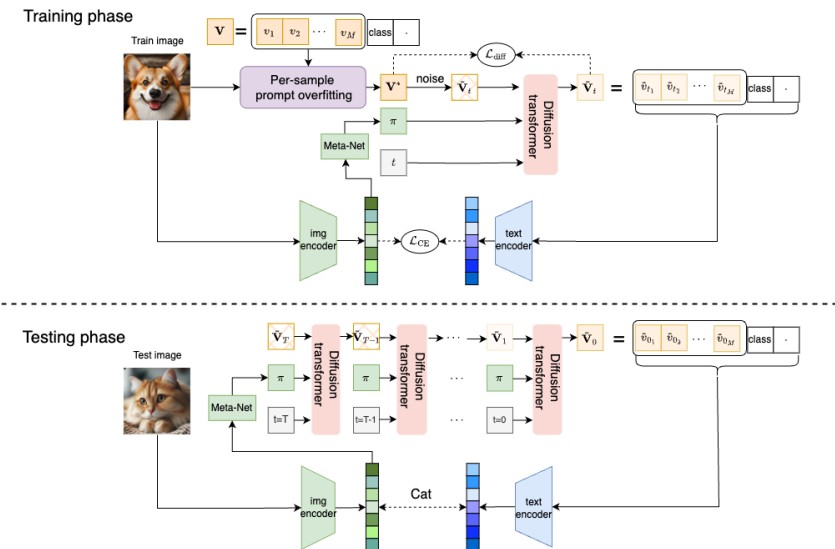

**Figure 3: Prompt diffusion.** (1) Training by generating prompts that are initially overfitted using per-sample overfitting. These prompts are then subjected to a noise injection before entering the forward diffusion process. The inputs for diffusion include noisy prompts $\tilde{V}_t^*$, the image features $\pi$, and a randomly chosen time step $t$, which leads to the generation of diffused prompts $\tilde{V}_t$. After training, the diffusion transformer can convert generic prompts into their overfitted counterparts for each sample. (2) During testing, the sampling process begins with an initial random noise $\tilde{V}_T$, which is gradually refined into diffused prompts $\tilde{V}_t$. At each time step $t$, the sampling process incorporates the previous state $\tilde{V}_{t-1}$, test image features $\pi$, and current time step $t$ as inputs. The resulting diffused prompts $\tilde{V}_0$ are then employed to make test sample predictions. Throughout $T$ with our diffusion transformer, the vanilla prompts are adapted into customized prompts that contain more specific information about the test sample, thereby enhancing prediction accuracy.

In the supplemental materials, we provide the computational graph, which showcases the sequential steps of the forward and inverse diffusion processes on the prompts. Our method balances adaptability and informativeness by incorporating probabilistic prompts with the diffusion model. Our model has also been applied to visual prompt tuning (VPT) (Jia et al., 2022) and multi-modal prompt learning (MaPLe, PromptSRC, and CoPrompt) (Khattak et al., 2023a;b; Roy & Etemad, 2024), generating visual prompts through a process identical to that used for generating text prompts.

**Testing phase.** During the testing phase, the generation of overfitted prompts is infeasible due to the unavailability of test sample labels. Consequently, the diffusion sampling process begins with the introduction of Gaussian noise $\tilde{V}_T$ alongside the computed image feature set $\pi$, followed by a systematic denoising procedure. To address an unseen test instance $x$, initial image feature computations $\pi$ are performed. After this, the noise vector $\epsilon$ is drawn from a standard normal distribution $\mathcal{N}(\mathbf{0}, \mathbf{I})$ for each model and data/set. This ensures a diverse starting point for each prompt without using multiple models, training multiple times, or employing different checkpoints. These elements, comprising $\tilde{V}_T$, $\pi$, and $\epsilon$, are then supplied to the trained prompt diffusion model to derive intermediate diffused prompts $\tilde{V}_{T-1}$, represented by $\tilde{V}_\theta(\tilde{V}_T, \pi, T)$. This iterative process unfolds over $T$ steps, culminating in the acquisition of the terminal diffused prompts $\tilde{V}_0 = \tilde{V}_\theta(\tilde{V}_1, \pi, t_0)$. Upon retrieval of $\tilde{V}_0$, integration with the text encoder occurs, facilitating the generation of relevant text features. The final stage involves the deployment of these features to predict the classification result for the test image, as delineated by Eq. (1). The diffusion sampling framework throughout the testing phase is shown at the bottom of Figure 3.

## 5 EXPERIMENTS

We validate the effectiveness of our approach across three widely adopted scenarios for evaluating prompt learning in vision-language models: (1) base-to-new generalization, (2) cross-dataset generalization, and (3) domain generalization.

**Table 1: Base-to-new generalization.** Prompts are derived from base class examples. The harmonic mean (H) underscores the trade-off in generalization. Top-performing results are emphasized in blue. By integrating our plugin within five different prompt learning methods, we consistently improve their average accuracy across 11 datasets, demonstrating enhanced performance over approaches without our plugin.

**(a) Average over 11 datasets.**

| | Base | New | H |
|---|---|---|---|
| VPT (Jia et al., 2022) | 72.53 | 72.34 | 72.43 |
| + Prompt Diffusion | 74.98 | 74.97 | 74.97 |
| CoCoOp (Zhou et al., 2022a) | 80.47 | 71.69 | 75.83 |
| + Prompt Diffusion | 81.35 | 74.97 | 78.02 |
| MaPLe (Khattak et al., 2023a) | 82.28 | 75.14 | 78.55 |
| + Prompt Diffusion | 83.39 | 77.32 | 80.24 |
| PromptSRC (Khattak et al., 2023b) | 84.26 | 76.10 | 79.97 |
| + Prompt Diffusion | 85.74 | 78.97 | 82.22 |
| CoPrompt (Roy & Etemad, 2024) | 84.00 | 77.23 | 80.48 |
| + Prompt Diffusion | 86.14 | 80.01 | 82.96 |

**(b) ImageNet.**

| | Base | New | H |
|---|---|---|---|
| VPT (Jia et al., 2022) | 74.45 | 69.22 | 71.74 |
| + Prompt Diffusion | 74.97 | 69.99 | 72.39 |
| CoCoOp (Zhou et al., 2022a) | 75.98 | 70.43 | 73.10 |
| + Prompt Diffusion | 76.46 | 70.97 | 73.61 |
| MaPLe (Khattak et al., 2023a) | 76.66 | 70.54 | 73.47 |
| + Prompt Diffusion | 77.01 | 71.03 | 73.89 |
| PromptSRC (Khattak et al., 2023b) | 77.60 | 70.73 | 74.01 |
| + Prompt Diffusion | 79.13 | 72.46 | 75.65 |
| CoPrompt (Roy & Etemad, 2024) | 77.67 | 71.27 | 74.33 |
| + Prompt Diffusion | 80.73 | 73.25 | 76.81 |

**(c) Caltech101.**

| | Base | New | H |
|---|---|---|---|
| VPT (Jia et al., 2022) | 96.92 | 93.44 | 95.15 |
| + Prompt Diffusion | 97.43 | 94.23 | 95.80 |
| CoCoOp (Zhou et al., 2022a) | 97.96 | 93.81 | 95.84 |
| + Prompt Diffusion | 98.12 | 94.97 | 96.52 |
| MaPLe (Khattak et al., 2023a) | 97.74 | 94.36 | 96.02 |
| + Prompt Diffusion | 97.25 | 95.98 | 96.61 |
| PromptSRC (Khattak et al., 2023b) | 98.10 | 94.03 | 96.02 |
| + Prompt Diffusion | 98.08 | 96.86 | 97.47 |
| CoPrompt (Roy & Etemad, 2024) | 98.27 | 94.90 | 96.55 |
| + Prompt Diffusion | 98.73 | 95.75 | 97.22 |

**(d) OxfordPets.**

| | Base | New | H |
|---|---|---|---|
| VPT (Jia et al., 2022) | 92.63 | 94.96 | 93.78 |
| + Prompt Diffusion | 93.17 | 97.18 | 95.14 |
| CoCoOp (Zhou et al., 2022a) | 95.20 | 97.69 | 96.43 |
| + Prompt Diffusion | 94.97 | 97.98 | 96.45 |
| MaPLe (Khattak et al., 2023a) | 95.43 | 97.76 | 96.58 |
| + Prompt Diffusion | 95.96 | 98.11 | 97.02 |
| PromptSRC (Khattak et al., 2023b) | 95.33 | 97.30 | 96.30 |
| + Prompt Diffusion | 95.44 | 98.05 | 96.73 |
| CoPrompt (Roy & Etemad, 2024) | 95.67 | 98.10 | 96.87 |
| + Prompt Diffusion | 96.74 | 98.91 | 97.81 |

**(e) StanfordCars.**

| | Base | New | H |
|---|---|---|---|
| VPT (Jia et al., 2022) | 65.06 | 74.68 | 69.54 |
| + Prompt Diffusion | 65.75 | 75.23 | 70.17 |
| CoCoOp (Zhou et al., 2022a) | 70.49 | 73.59 | 72.01 |
| + Prompt Diffusion | 70.98 | 75.32 | 73.08 |
| MaPLe (Khattak et al., 2023a) | 72.94 | 74.00 | 73.47 |
| + Prompt Diffusion | 73.11 | 75.03 | 74.06 |
| PromptSRC (Khattak et al., 2023b) | 78.27 | 74.97 | 76.58 |
| + Prompt Diffusion | 80.14 | 76.15 | 78.09 |
| CoPrompt (Roy & Etemad, 2024) | 76.97 | 74.40 | 75.66 |
| + Prompt Diffusion | 79.13 | 75.83 | 77.44 |

**(f) Flowers102.**

| | Base | New | H |
|---|---|---|---|
| VPT (Jia et al., 2022) | 76.23 | 71.55 | 73.82 |
| + Prompt Diffusion | 77.29 | 72.33 | 74.73 |
| CoCoOp (Zhou et al., 2022a) | 94.87 | 71.75 | 81.71 |
| + Prompt Diffusion | 94.17 | 75.73 | 83.95 |
| MaPLe (Khattak et al., 2023a) | 95.92 | 72.46 | 82.56 |
| + Prompt Diffusion | 95.90 | 73.14 | 82.99 |
| PromptSRC (Khattak et al., 2023b) | 98.07 | 76.50 | 85.95 |
| + Prompt Diffusion | 98.96 | 78.27 | 87.41 |
| CoPrompt (Roy & Etemad, 2024) | 97.27 | 76.60 | 85.71 |
| + Prompt Diffusion | 98.73 | 78.49 | 87.45 |

**(g) Food101.**

| | Base | New | H |
|---|---|---|---|
| VPT (Jia et al., 2022) | 89.27 | 90.50 | 89.88 |
| + Prompt Diffusion | 89.97 | 92.12 | 91.03 |
| CoCoOp (Zhou et al., 2022a) | 90.70 | 91.29 | 90.99 |
| + Prompt Diffusion | 90.21 | 92.01 | 91.10 |
| MaPLe (Khattak et al., 2023a) | 90.71 | 92.05 | 91.38 |
| + Prompt Diffusion | 91.26 | 93.11 | 92.18 |
| PromptSRC (Khattak et al., 2023b) | 90.67 | 91.53 | 91.10 |
| + Prompt Diffusion | 90.74 | 92.58 | 91.65 |
| CoPrompt (Roy & Etemad, 2024) | 90.73 | 92.07 | 91.40 |
| + Prompt Diffusion | 91.34 | 92.98 | 91.25 |

**(h) FGVCAircraft.**

| | Base | New | H |
|---|---|---|---|
| VPT (Jia et al., 2022) | 28.23 | 32.21 | 30.09 |
| + Prompt Diffusion | 28.82 | 35.07 | 31.64 |
| CoCoOp (Zhou et al., 2022a) | 33.41 | 23.71 | 27.74 |
| + Prompt Diffusion | 34.21 | 35.27 | 34.73 |
| MaPLe (Khattak et al., 2023a) | 37.44 | 35.61 | 36.50 |
| + Prompt Diffusion | 37.11 | 36.15 | 36.62 |
| PromptSRC (Khattak et al., 2023b) | 42.73 | 37.87 | 40.15 |
| + Prompt Diffusion | 44.81 | 39.98 | 42.26 |
| CoPrompt (Roy & Etemad, 2024) | 40.20 | 39.33 | 39.76 |
| + Prompt Diffusion | 42.35 | 41.27 | 41.80 |

**(i) SUN397.**

| | Base | New | H |
|---|---|---|---|
| VPT (Jia et al., 2022) | 75.14 | 76.89 | 76.00 |
| + Prompt Diffusion | 75.74 | 77.82 | 76.77 |
| CoCoOp (Zhou et al., 2022a) | 79.74 | 76.86 | 78.27 |
| + Prompt Diffusion | 80.14 | 77.53 | 78.81 |
| MaPLe (Khattak et al., 2023a) | 80.82 | 78.70 | 79.75 |
| + Prompt Diffusion | 81.03 | 79.54 | 80.28 |
| PromptSRC (Khattak et al., 2023b) | 82.67 | 78.47 | 80.52 |
| + Prompt Diffusion | 84.15 | 80.27 | 82.16 |
| CoPrompt (Roy & Etemad, 2024) | 82.63 | 80.03 | 81.31 |
| + Prompt Diffusion | 84.71 | 81.97 | 83.32 |

**(j) DTD.**

| | Base | New | H |
|---|---|---|---|
| VPT (Jia et al., 2022) | 56.71 | 57.25 | 56.98 |
| + Prompt Diffusion | 58.43 | 58.13 | 58.28 |
| CoCoOp (Zhou et al., 2022a) | 77.01 | 56.00 | 64.85 |
| + Prompt Diffusion | 73.43 | 60.19 | 66.15 |
| MaPLe (Khattak et al., 2023a) | 80.36 | 59.18 | 68.16 |
| + Prompt Diffusion | 80.25 | 59.44 | 68.62 |
| PromptSRC (Khattak et al., 2023b) | 83.37 | 62.97 | 71.75 |
| + Prompt Diffusion | 85.71 | 65.07 | 73.98 |
| CoPrompt (Roy & Etemad, 2024) | 83.13 | 64.73 | 72.79 |
| + Prompt Diffusion | 85.14 | 65.96 | 74.33 |

**(k) EuroSAT.**

| | Base | New | H |
|---|---|---|---|
| VPT (Jia et al., 2022) | 67.57 | 59.69 | 63.39 |
| + Prompt Diffusion | 67.26 | 69.01 | 68.13 |
| CoCoOp (Zhou et al., 2022a) | 87.49 | 60.04 | 71.21 |
| + Prompt Diffusion | 88.13 | 70.22 | 78.16 |
| MaPLe (Khattak et al., 2023a) | 94.07 | 73.23 | 82.35 |
| + Prompt Diffusion | 94.76 | 73.34 | 82.69 |
| PromptSRC (Khattak et al., 2023b) | 92.90 | 73.90 | 82.32 |
| + Prompt Diffusion | 93.94 | 76.07 | 84.07 |
| CoPrompt (Roy & Etemad, 2024) | 94.60 | 78.57 | 85.84 |
| + Prompt Diffusion | 94.98 | 80.17 | 86.95 |

**(l) UCF101.**

| | Base | New | H |
|---|---|---|---|
| VPT (Jia et al., 2022) | 75.65 | 75.31 | 75.48 |
| + Prompt Diffusion | 76.31 | 76.23 | 76.27 |
| CoCoOp (Zhou et al., 2022a) | 82.33 | 73.45 | 77.64 |
| + Prompt Diffusion | 81.97 | 77.03 | 79.42 |
| MaPLe (Khattak et al., 2023a) | 83.00 | 78.66 | 80.77 |
| + Prompt Diffusion | 82.86 | 79.64 | 81.22 |
| PromptSRC (Khattak et al., 2023b) | 87.10 | 78.80 | 82.74 |
| + Prompt Diffusion | 88.21 | 79.91 | 83.86 |
| CoPrompt (Roy & Etemad, 2024) | 86.90 | 79.57 | 83.07 |
| + Prompt Diffusion | 88.14 | 80.28 | 84.03 |

## 5.1 EXPERIMENTAL SETUP

**15 diverse datasets.** For base-to-new generalization and cross-dataset gneralization, we follow CLIP (Radford et al., 2021a) and CoOp (Zhou et al., 2021) to use 11 image classification datasets, *i.e.*, ImageNet (Deng et al., 2009) and Caltech101 (Fei-Fei et al., 2004) for generic object classification, OxfordPets (Parkhi et al., 2012), StanfordCars (Krause et al., 2013), Flowers102 (Nilsback & Zisserman, 2008), Food101 (Bossard et al., 2014) and FGVCAircraft (Maji et al., 2013) for fine-grained image recognition, EuroSAT (Helber et al., 2019) for satellite image classification, UCF101 (Soomro et al., 2012) for action classification, DTD (Cimpoi et al., 2014) for texture classification, and SUN397 (Xiao et al., 2010) for scene recognition. For domain generalization, we follow CoOp (Zhou et al., 2021) with ImageNet as the source dataset, and we select four variants of ImageNet: ImageNetV2 (Recht et al., 2019), ImageNet-Sketch (Wang et al., 2019), ImageNet-A (Hendrycks et al., 2021b) and ImageNet-R (Hendrycks et al., 2021a) as the target datasets.

**5 prompt learning baselines.** For comparative evaluation, we employ several established baselines: (1) Textual prompt learning CoCoOp (Zhou et al., 2022a); (2) Visual prompt tuning (VPT) (Jia et al., 2022), representing the visual prompt learning method; (3) Multi-modal prompt learning (MaPLe (Khattak et al., 2023a), PromptSRC (Khattak et al., 2023b)), and CoPrompt (Roy & Etemad, 2024) employing prompt learning in both the visual and textual domains. Note that our method acts as a plugin that is easily integrated into each of these methods.

**Training details.** To ensure a fair comparison, we utilize the CLIP-ViT-B/16 as the base pre-training model for CoCoOp (Zhou et al., 2022a), and VPT (Jia et al., 2022), setting the prompt token count to 4. This configuration is based on recommendations in (Zhou et al., 2022a), indicating

**Table 2: Cross-dataset generalization.** Accuracy (%) evaluation for prompts learned from the source dataset. Our plugin consistently enhances existing prompt learning methods, whether textual, visual, or multi-modal.

| | Source | Target | | | | | | | | | | | |
| | ImageNet | Caltech101 | OxfordPets | StanfordCars | Flowers102 | Food101 | Aircraft | SUN397 | DTD | EuroSAT | UCF101 | Average |
|---|---|---|---|---|---|---|---|---|---|---|---|---|
| VPT (Jia et al., 2022) | 68.92 | 93.07 | 89.44 | 64.77 | 67.79 | 84.91 | 23.72 | 66.16 | 45.02 | 37.74 | 67.00 | 63.96 |
| + Prompt diffusion | 70.23 | 94.71 | 90.93 | 65.53 | 68.93 | 85.71 | 24.81 | 66.98 | 46.16 | 39.67 | 67.91 | 65.11 |
| CoCoOp (Zhou et al., 2022a) | 71.02 | 94.43 | 90.14 | 65.32 | 71.88 | 86.06 | 22.94 | 67.36 | 45.73 | 45.37 | 68.21 | 65.74 |
| + Prompt diffusion | 71.98 | 95.07 | 91.11 | 66.73 | 73.52 | 87.18 | 22.23 | 68.25 | 46.84 | 47.13 | 69.53 | 66.76 |
| MaPLe (Khattak et al., 2023a) | 70.72 | 93.53 | 90.49 | 65.57 | 72.23 | 86.20 | 24.74 | 67.01 | 46.49 | 48.06 | 68.69 | 66.30 |
| + Prompt diffusion | 71.23 | 95.98 | 92.49 | 67.17 | 74.13 | 88.24 | 26.23 | 69.43 | 47.95 | 49.73 | 69.53 | 68.09 |
| PromptSRC (Khattak et al., 2023b) | 71.27 | 93.60 | 90.25 | 65.70 | 70.25 | 86.15 | 23.90 | 67.10 | 46.87 | 45.50 | 68.75 | 65.81 |
| + Prompt diffusion | 71.73 | 96.01 | 93.13 | 68.12 | 73.71 | 88.31 | 26.14 | 70.21 | 48.35 | 48.15 | 70.24 | 68.23 |
| CoPrompt (Roy & Etemad, 2024) | 70.80 | 94.50 | 90.73 | 65.67 | 72.30 | 86.43 | 24.00 | 67.57 | 47.07 | 51.90 | 69.73 | 67.00 |
| + Prompt diffusion | 71.46 | 96.12 | 93.94 | 68.81 | 74.98 | 88.11 | 26.31 | 71.73 | 49.15 | 54.41 | 71.14 | 69.47 |

optimal performance with a more concise context length. For MaPLe (Khattak et al., 2023a), PromptSRC (Khattak et al., 2023b) and CoPrompt (Roy & Etemad, 2024), the prompt depth $M$ is adjusted to 9, and we configure the language and vision prompt lengths at two tokens each. In the diffusion preprocessing stage, we adapt the strategy of positional token assignment (Dosovitskiy et al., 2021) to the input prompts $\tilde{V}^*$ and the image features $\pi$. Furthermore, the diffusion time step $t$ is encoded as a series of individual tokens, adopting a frequency-based vector representation scheme (Mildenhall et al., 2021). We set the diffusion time step $t$ as 100 for our experiments. Our transformer-based model architecture is the same as the GPT-2 framework (Radford et al., 2019). This includes a 12-layer transformer, a linear transformation, and an attention mechanism with 16 heads. The batch size is 32 for all prompt-based models, except for CoCoOp, which is trained with a batch size of 4. Each model leverages a learning rate 0.0035 applied through the SGD optimizer on a single NVIDIA A100 GPU for execution. Code will be made available.

**Evaluation setting.** Across all experiments, we benchmark the models' performance in a 16-shot setting, standardizing the number of training epochs to 50 for each baseline and dataset. The appendix presents a 4-shot experiment, compares outcomes across different epochs, and evaluates various parameter-efficient approaches. For consistency, all results from learning-based methods are computed as an average over three random seeds.

## 5.2 COMPARATIVE EXPERIMENTS

**Base-to-new generalization.** Table 1 shows that various prompting methods, when combined with our prompt diffusion approach, consistently surpass the average performance across all datasets. Regarding base class accuracy averaged across 11 datasets, our approach advances VPT, CoCoOp, MaPle, PromptSRC, and CoPrompt by 2.54%, 1.08%, 1.11%, 2.25% and 2.48%, respectively, showcasing that our approach strengthens the adaptation of existing methods. When it comes to recognizing new classes, our approach also shows good improvement, with gains of 2.60% for VPT, 1.26% for CoCoOp, 2.78% for MaPle 1.81% for PromptSRC, and 2.87% for CoPrompt, emphasizing its effectiveness in dealing with unseen samples. Regarding the harmonic mean, which considers both base and new classes, our method retains a superior few-shot generalization capacity across all datasets compared to baseline models. Notably, CoPrompt, with our prompt diffusion, consistently outperforms all other methods across most datasets, demonstrating the advantages of using both modalities in prompt learning. Our prompt diffusion, applied to different prompt learning models, consistently improves the outcomes by generating more informative and precise prompts.

**Cross-dataset generalization.** Our study assesses how well models can adapt prompt learning from one dataset and apply it effectively to different datasets for cross-dataset generalization. We test the zero-shot transfer capabilities of the models on a wide range of 10 datasets. As shown in Table 2, our prompt diffusion substantially improves the average transfer performance of models like VPT, CoCoOp, MaPLe, PromptSRC and CoPrompt, with respective increases of 1.15%, 1.02%, 1.79%, 2.43%, and 2.47%. These results not only confirm the effectiveness of our method in enhancing cross-dataset generalization but also highlight its versatility across various prompt learning methods.

**Domain generalization.** The performance of various ImageNet variants, which have a domain shift compared with the source dataset, is evaluated. Table 3 summarizes these findings, highlighting not only the improvement in performance across VPT, CoCoOp, MaPLe, PromptSRC, and CoPrompt but also the maintenance of performance on the source dataset itself. Interest-

ingly, using prompt diffusion with CoCoOp performs better than all multi-modal prompt learning methods on the ImageNet-A dataset. This could be due to CoCoOp's emphasis on textual prompt learning, which may be more suited to these types of datasets. ImageNet-A images frequently display anomalies or atypical features, whereas visual prompts could highlight deceptive or overly intricate aspects, making classification less accurate. Thus, when addressing real-world datasets like ImageNet-A, it is advantageous to use textual prompting with our prompt diffusion.

Since such datasets often contain natural images, textual prompts can leverage the semantic context effectively. On the other hand, in scenarios involving a clear distribution shift (*e.g. sketch, cartoon*), employing a multi-modality prompt with our prompt diffusion is more effective. From the results of these experiments, our method fosters a level of adaptability that allows models to maintain their initial generalizability even after being fine-tuned to limited datasets.

Table 3: **Domain generalization.** Accuracy (%) evaluation on target datasets using prompts learned from a source dataset. Our method delivers consistent, prompt learning improvements across all datasets.

| | Source | Target | | | | |
|---|---|---|---|---|---|---|
| | ImageNet | -V2 | -S | -A | -R | Average |
| VPT (Jia et al., 2022) | 68.92 | 61.84 | 47.64 | 46.50 | 75.86 | 57.96 |
| + Prompt diffusion | 70.23 | 62.97 | 48.77 | 47.25 | 77.06 | 59.01 |
| CoCoOp (Zhou et al., 2022a) | 71.02 | 64.07 | 48.75 | 50.63 | 76.18 | 59.91 |
| + Prompt diffusion | 71.98 | 65.28 | 50.11 | 52.23 | 77.50 | 61.25 |
| MaPLe (Khattak et al., 2023a) | 70.72 | 64.07 | 49.15 | 50.90 | 76.98 | 60.83 |
| + Prompt diffusion | 71.23 | 65.49 | 50.46 | 52.18 | 78.31 | 62.36 |
| PromptSRC (Khattak et al., 2023b) | 71.27 | 64.35 | 49.55 | 50.90 | 77.80 | 60.65 |
| + Prompt diffusion | 71.73 | 66.33 | 51.21 | 52.02 | 79.86 | 62.88 |
| CoPrompt (Roy & Etemad, 2024) | 70.80 | 64.25 | 49.43 | 50.50 | 77.51 | 60.42 |
| + Prompt diffusion | 71.46 | 66.01 | 50.71 | 51.75 | 80.76 | 62.30 |

## 5.3 Ablation experiments

**Benefit of the diffusion model.** To confirm that the performance gain of our model can be attributed to the diffusion model, we first conducted the experiments using MLP and transformers as non-generative models, using overfitted prompts as supervision. We also compared it with three widely used generative models: generative adversarial networks (GAN) (Goodfellow et al., 2020), variational auto-encoders (VAE) (Kingma & Welling, 2013), and normalizing flows (Rezende & Mohamed, 2015). First, we obtain the overfitted prompts with per-sample prompt overfitting. In the case of non-generative models, the process involves solely using image features $\pi$, and then employing overfitted prompts, $V^*$, for supervising the training of MLP and transformer models. The input is extended for GANs, VAE and normalizing flows to include image features $\pi$ and a variable $\epsilon$ sampled from a standard normal distribution $\mathcal{N}(0, I)$. These inputs, along with the overfitted prompts $V^*$, are used to supervise and train these three types of generative models. Table 4 shows the diffusion model outperforms all variants in accuracy. Specifically, our proposed per-sample overfitting integration with MLP and transformer architectures shows a slight harmonic mean improvement over the CoCoOp baseline, validating the effectiveness of our per-sample prompt overfitting. Notably, our diffusion model presents a good increase in accuracy, surpassing the GAN, VAE and normalizing flows models by 2.01%, 1.85% and 1.70%, respectively.

Table 4: **Benefit of diffusion model** in the base-to-new generalization.

| | Base | New | H |
|---|---|---|---|
| CoCoOp (Zhou et al., 2022a) | 80.47 | 71.69 | 75.83 |
| w/ MLP | 79.18 | 71.98 | 75.41 |
| w/ Transformer | 80.17 | 72.03 | 75.88 |
| w/ GAN | 81.15 | 71.44 | 75.99 |
| w/ VAE | 80.73 | 72.09 | 76.17 |
| w/ Normalizing flows | 80.65 | 72.43 | 76.32 |
| **w/ Diffusion** | 81.35 | 74.97 | 78.02 |

**Effect of the number of function evaluation.** Our prompt diffusion utilizes the fast ODE-based sampling strategy introduced by (Zhou et al., 2024) enabling efficient sampling with a reduced number of timesteps during testing. In Figure 4, we analyze the effect of different numbers of function evaluations (NFE) on both final performance and inference time. Our findings indicate that at an NFE of 5, our method achieves the best balance between performance and prediction time. In comparison to the original CoCoOp, our approach results in only a 0.045-second increase in prediction time while delivering a substantial performance improvement. This highlights the effectiveness of our method in balancing accuracy and computational efficiency.

**Impact of iterations on per-sample prompt overfitting.** In our prompt diffusion method, per-sample prompt overfitting is crucial to generate optimal prompts during training. Figure 5 shows that as iterations increase, accuracy on novel classes improves for all methods, peaking at iteration 5. This shows that the quality of the optimal prompt directly influences the final performance. Moreover, our

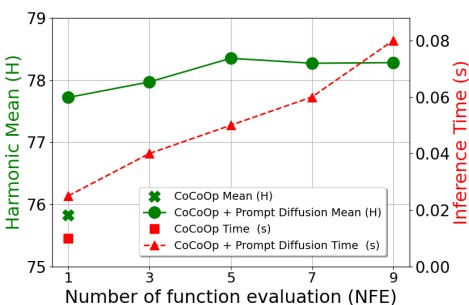

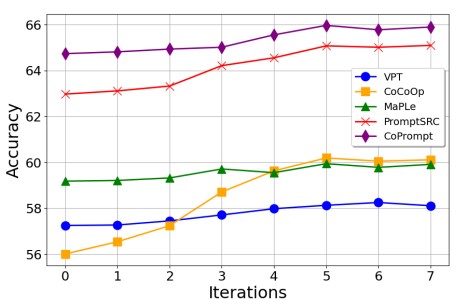

**Figure 4: Effect of number of function evaluation** on base-to-new generalization.

**Figure 5: Impact of iterations** on per-sample prompt overfitting for novel classes.

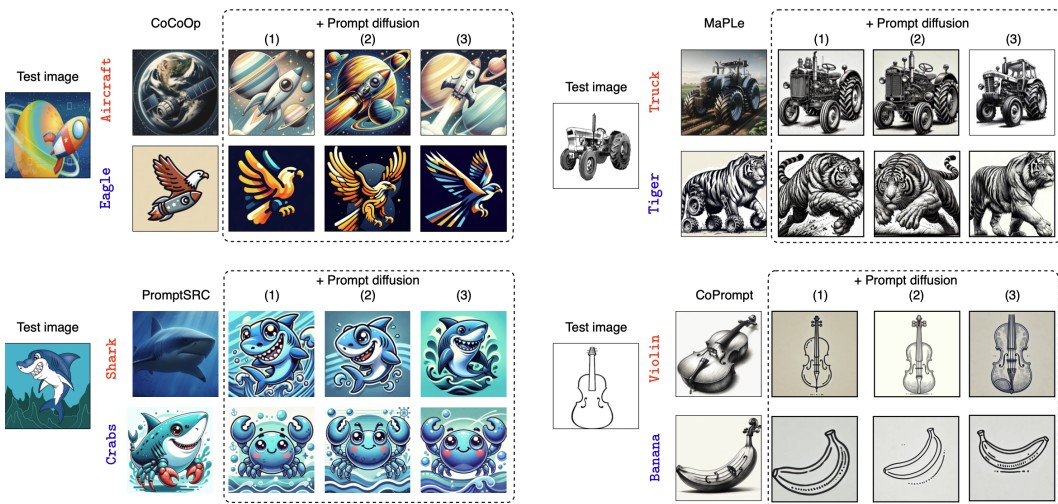

**Figure 6: Visualization of generated prompts** by ControlNet (Zhang et al., 2023). We generate diverse images by utilizing three distinct Monte Carlo prompt samples, each derived from our prompt distribution and based on varying random noise. When using ground truth names (`red` names), images produced by CoCoOp, MaPLe, PromptSRC, and CoPrompt are more realistic, while prompt diffusion incorporates domain-specific details from the test image. Regarding the other classes (`blue` names), CoCoOp, MaPLe, PromptSRC, and CoPrompt blend true class features with others, potentially leading to confusion, but our plugin using these methods can generate a stylized version of the specified class. This suggests that our plugin enables the distilling of unique domain details from the test image without conflating them with class labels.

prompt diffusion effectively learns the transformation from a vanilla prompt to an optimal prompt throughout training using a diffusion transformer. As a result, during testing, our method can generate a sample-specific prompt for each test sample, thereby improving accuracy.

**Visualization of generated prompts.** We also visualize the generated per-sample prompts during inference in Figure 6, demonstrating our diffusion prompting method effectively distills unique domain details from the test image without mixing them with class labels. This shows the better capability of the diffusion model in refining the prompt learning process for vision-language tasks.

## 6 CONCLUSION

Our approach addresses the limitations of fixed prompts by introducing a method that crafts customized prompts for individual test samples, enhancing model robustness against distributional shifts. The diffusion model serves as the backbone of this method, enabling a generative process that refines prompts from a random initialization to an optimized state, tailored to each specific instance. The versatility and modality-agnostic nature of prompt diffusion mark it as an universally applicable solution that integrates smoothly with existing prompt learning methods, regardless of the data type. The empirical results across a wide range of datasets validate the efficacy of our method, demonstrating its increased robustness in generalization tasks.

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

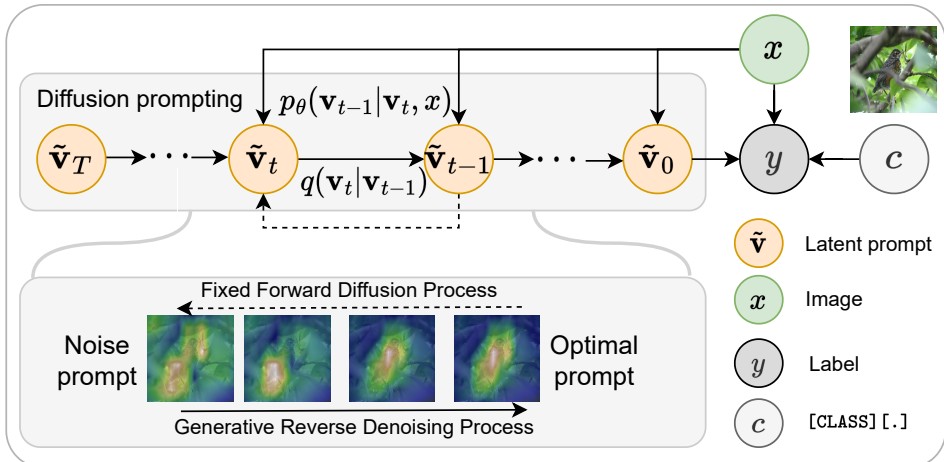

Figure 7: **Computational Graph and Diffused Prompt**. The top diagram illustrates the computational structure of our method. On the bottom left, we showcase the graphical representation of our method's diffused prompt. Through diffusion sampling techniques, the diffused prompt $\mathbf{V}_{t-1}$ emerges as a fusion of $\mathbf{V}_t$ and $x$. The resultant prediction $y$ is then formed by leveraging the diffused prompt, expanded as "[CLASS]" label $\mathbf{c}: \{\mathbf{V}_0, \mathbf{c}\}$, and paired with the image descriptor $x$. Within the shaded rectangle, dashed arrows denote the diffusion procedure, while solid arrows highlight the sampling steps.

## A COMPUTATIONAL GRAPH OF DIFFUSION PROMPTING

In this section, we illustrate the computational graph of diffusion prompting in Figure 7. The figure is divided into two parts: the top diagram displays the overall computational structure of the method, while the bottom left part presents a graphical representation of the method's diffused prompt. In this process, the diffused prompt $\mathbf{V}_{t-1}$ is created through a fusion of $\mathbf{V}_t$ and the image descriptor $x$ using diffusion sampling techniques. This results in the prediction $y$, which is generated by combining the diffused prompt, expanded as the "[CLASS]" label $\mathbf{c}: \{\mathbf{V}_0, \mathbf{c}\}$, with the image descriptor $x$. The shaded rectangle in the diagram helps to visually differentiate the components of the process, where dashed arrows indicate the diffusion steps, and solid arrows represent the sampling stages. This figure provides a clear and concise visual representation of the complex processes involved in diffused prompting, highlighting the intricate interactions between different components of the computational model.

## B ADDITIONAL RELATED WORKS

**Diffusion models.** This class of neural generative models is characterized by the employment of stochastic diffusion processes akin to those observed in thermodynamic systems (Sohl-Dickstein et al., 2015; Song et al., 2020). The operational principle of these models involves a sequential noise addition to data samples, followed by a learned neural network's effort to reverse this process. This is achieved by gradually denoising the noise-saturated sample to retrieve data reflecting the trained data distribution. Significant strides in the realm of image generation have been accredited to the works of Ho *et al.* (Ho et al., 2020) and Song *et al.* (Song et al., 2020), while Dhariwal and Nichol (Dhariwal & Nichol, 2021) have been pivotal in pioneering classifier-guided diffusion for generation under specific conditions. Building on this foundation, GLIDE (Nichol et al., 2021) has further refined the methodology by incorporating conditioning on textual representations derived from CLIP. The concept of classifier-free guidance introduced by Ho *et al.* (Ho & Salimans, 2022) has brought forward a method of conditioning that judiciously balances fidelity and diversity, leading to notable enhancements in model performance (Nichol et al., 2021). However, guided diffusion models typically necessitate an extensive corpus of image-annotation pairs for effective training, prompting Hu *et al.* (Hu et al., 2023) to suggest the novel concept of self-guided diffusion models. More contemporary developments include Hyperdiffusion (Lutati & Wolf, 2022; Erkoç et al., 2023), which targets the generation of implicit neural representations and 3D reconstruction through diffusion in weight space. To the best of our knowledge, we are the first to introduce diffusion models into

**Table 5:** Comparison with CoOp for various epochs.

| Epochs | Base | New | H |
|--------|-------|-------|-------|
| 10 | 80.29 | 73.51 | 76.26 |
| 50 | 81.35 | 74.97 | 77.72 |
| 100 | 81.47 | 74.88 | 77.70 |
| 200 | 81.78 | 74.24 | 77.65 |

**Table 6:** Comparison with 4-shots on domain generalization. Our results are competitive for all domains.

| | Source | Target | | | | |
|---|---|---|---|---|---|---|
| | ImageNet | -V2 | -S | -A | -R | Average |
| VPT (Jia et al., 2022) | 69.24 | 62.13 | 45.78 | 48.16 | 72.91 | 57.37 |
| + Prompt Diffusion | 70.27 | 63.83 | 48.15 | 50.97 | 76.15 | 60.12 |
| CoCoOp (Zhou et al., 2022a) | 70.13 | 63.05 | 46.48 | 49.36 | 73.80 | 58.17 |
| + Prompt Diffusion | 70.96 | 64.12 | 48.92 | 51.47 | 76.93 | 60.75 |
| MaPLe (Khattak et al., 2023a) | 70.72 | 64.07 | 49.15 | 50.90 | 76.98 | 60.83 |
| + Prompt Diffusion | 71.23 | 65.24 | 50.21 | 51.93 | 78.06 | 62.11 |

the realm of prompt learning. Our diffusion prompting involves gradually refining prompts with a diffusion transformer, which leads to the development of custom prompts tailored to each sample, thereby enhancing the accuracy of predictions and their generalization across downstream tasks.

## C  HYPERPARAMETER SENSITIVITY AND FEW-SHOT.

Table 5 presents a comparison of our model's performance over different epochs relative to CoOp's training duration of 200 epochs. Our model reaches convergence around the 50-epoch mark and surpasses the performance of CoOp after 200 epochs. Additionally, we also conduct a few-shot learning experiment (4-shot) similar to those conducted with CoOp and CoCoOp, as shown in Table 6. In these comparisons, our model consistently achieves improved performance across a range of datasets.

## D  PARAMETER-EFFICIENT COMPARISON.

Table 7 contrasts our approach with four other parameter-efficient fine-tuning techniques. Our integration with MaPLe (Khattak et al., 2023a) showcases superior average performance, underscoring its superior ability to generalize in comparison to other parameter-efficient fine-tuning approaches. Furthermore, we have applied our plugin in conjunction with LLU (Ibing et al., 2023) in a base-to-new setting, where it also exhibits enhanced performance relative to LLU alone.

## E  EFFECT OF PROMPT LENGTH ON PERFORMANCE

The length of prompts plays a significant role in the final performance of prompt learning methods. To analyze the impact of prompt length, we conducted experiments with our prompt diffusion method using different prompt lengths across various baseline methods. It is worth noting that the prompt lengths used in our experiments align with the default prompt lengths adopted by the respective baseline methods: $L = 4$ for VPT and CoCoOp, and $L = 9$ for MaPLe. The results are summarized in Tables 8, 9, and 10. These results demonstrate that the optimal prompt length varies across different baselines, with moderate lengths generally leading to better performance. For our method, we use the respective default prompt lengths for fair comparisons: $L = 4$ for VPT and CoCoOp, and $L = 9$ for MaPLe. This ensures consistency and fairness in our evaluation.

**Table 7:** Comparison with parameter-efficient fine-tuning methods in the base-to-new setting across 11 datasets.

|  | Venues | Base | New | H |
|---|---|---|---|---|
| ProGrad (Zhu et al., 2023) | ICCV 23 | 82.79 | 68.55 | 74.46 |
| CLIP Adapter (Gao et al., 2024) | IJCV | 82.62 | 70.97 | 76.02 |
| LLU (Ibing et al., 2023) | CVPR 23 | 83.48 | 74.47 | 78.46 |
| MaPLe (Khattak et al., 2023a) | CVPR 23 | 82.28 | 75.14 | 78.55 |
| LLU + Diffusion Prompt | | 84.45 | 75.99 | 79.17 |
| MaPLe + Diffusion Prompt | | 83.39 | 77.12 | 79.96 |

**Table 8:** Effect of prompt length on VPT + Prompt Diffusion.

| Length ($L$) | Base | New | H |
|---|---|---|---|
| 4 | 74.98 | 74.97 | 74.97 |
| 8 | 75.73 | 75.26 | 75.49 |
| 16 | 72.97 | 72.65 | 72.80 |

**Table 9:** Effect of prompt length on CoCoOp + Prompt Diffusion.

| Length ($L$) | Base | New | H |
|---|---|---|---|
| 4 | 81.35 | 74.97 | 78.02 |
| 8 | 82.97 | 76.93 | 79.84 |
| 16 | 78.91 | 74.11 | 76.43 |

**Table 10:** Effect of prompt length on MaPLe + Prompt Diffusion.

| Length ($L$) | Base | New | H |
|---|---|---|---|
| 4 | 82.93 | 76.15 | 79.40 |
| 9 | 83.39 | 77.32 | 80.24 |
| 16 | 82.77 | 75.93 | 79.20 |

# F    COMPUTATIONAL LOAD AND TRAINING EFFICIENCY

To address concerns about the computational load introduced by our method, we conducted a comparative analysis of training time across baseline methods and our proposed approach. While our method introduces a slightly higher computational load due to the per-sample prompt overfitting step and the diffusion process, the increase is modest. Specifically, the per-sample prompt overfitting step requires only three iterations to generate the overfitted prompts, ensuring efficiency without compromising performance. The training times (in hours) and corresponding performance (Base, New, and Harmonic Mean) for each method are summarized in Table 11. The results demonstrate that while the training time increases slightly (approximately $1.1\times$ to $1.3\times$ that of baseline methods), our method consistently achieves better performance in terms of Base, New, and Harmonic Mean (H). This highlights a favorable trade-off between computational load and performance. The per-sample prompt overfitting step, coupled with the diffusion model, plays a critical role in enhancing the model's adaptability to diverse samples.

# G    ADDITIONAL EXPERIMENTS ON VPT-DEEP

To demonstrate the versatility of our method, we conducted additional experiments with VPT-deep. The results, presented in Table 12, show that incorporating our prompt diffusion significantly improves the performance of VPT-deep across all metrics, including Base, New, and Harmonic Mean (H). These results confirm that our approach is not limited to VPT-shallow but can also effectively enhance

**Table 11:** Training time (in hours) and performance comparison across baseline methods and our approach.

| Method | Base | New | H | Training Time (hours) |
|---|---|---|---|---|
| VPT Jia et al. (2022) | 72.53 | 72.34 | 72.43 | 25 |
| + Prompt Diffusion | 74.98 | 74.97 | 74.97 | 28 |
| CoCoOp Zhou et al. (2022a) | 80.47 | 71.69 | 75.83 | 17 |
| + Prompt Diffusion | 81.35 | 74.97 | 78.02 | 20 |
| MaPLe Khattak et al. (2023a) | 82.28 | 75.14 | 78.55 | 21 |
| + Prompt Diffusion | 83.39 | 77.32 | 80.24 | 27 |
| PromptSRC Khattak et al. (2023b) | 84.26 | 76.10 | 79.97 | 23 |
| + Prompt Diffusion | 85.74 | 78.97 | 82.22 | 30 |
| CoPrompt Roy & Etemad (2024) | 84.00 | 77.23 | 80.48 | 23 |
| + Prompt Diffusion | 86.14 | 80.01 | 82.96 | 30 |

the VPT-deep prompt learning paradigm. The consistent improvement across all metrics highlights the adaptability and effectiveness of our method in different prompt learning settings.

**Table 12:** Performance comparison of VPT-deep with and without Prompt Diffusion.

| Method | Base | New | H |
|---|---|---|---|
| VPT-deep | 74.15 | 74.01 | 74.08 |
| + Prompt Diffusion | 77.15 | 76.89 | 77.02 |

# H  PROMPT DIFFUSION FOR VARIOUS PROMPT LEARNING METHODS

While the proposed method leverages the meta-net $\pi$ in CoCoOp, it is fully adaptable to other prompt learning methods, such as VPT and MaPLe, which do not rely on $\pi$. For these methods, during training, we perform **per-sample prompt overfitting** to generate the corresponding overfitted prompts or tokens. These overfitted prompts or tokens are then reconstructed using the diffusion process. Specifically, for multi-modal prompt learning methods, such as MaPLe, we generate overfitted prompts for both the textual and visual branches during the per-sample prompt overfitting stage. The diffusion process then reconstructs these overfitted prompts independently for each modality. This dual reconstruction ensures that both the textual and visual prompts are refined and aligned with their respective input modalities, contributing to improved performance in multi-modal tasks. Finally, the reconstructed prompts or tokens are embedded back into the original models, such as VPT and MaPLe, for prediction. This flexibility highlights that our method is not tied to any specific architecture. Instead, it serves as a **plug-and-play module** that can seamlessly integrate with various prompt learning paradigms, including visual prompt tuning and multi-modal prompt tuning. By adapting to the needs of different frameworks, our method enhances generalizability and improves performance across a wide range of tasks.

# I  EFFECT OF LOSS WEIGHT $\beta$ ON PERFORMANCE

To analyze the impact of the loss weight $\beta$ in Equation (8), we conducted experiments with different values of $\beta$ across VPT, CoCoOp, and MaPLe. The results are summarized in Tables 13, 14, and 15. The results show that the best performance across all metrics (Base, New, and Harmonic Mean) is achieved when $\beta = 0.01$. This indicates that a small weight for the diffusion loss term provides the optimal balance between the cross-entropy loss and the reconstruction objective. Setting $\beta$ too high ($\beta = 1$) places excessive emphasis on the diffusion term, slightly degrading performance. Conversely, setting $\beta = 0$ removes the benefits of the diffusion process entirely, leading to a significant drop in performance.

**Table 13:** Effect of $\beta$ on VPT + Prompt Diffusion.

| $\beta$ | Base | New | H |
|---|---|---|---|
| 0 | 72.53 | 72.34 | 72.43 |
| 0.01 | 74.98 | 74.97 | 74.97 |
| 0.1 | 73.45 | 74.71 | 74.07 |
| 1 | 73.16 | 73.15 | 73.16 |

**Table 14:** Effect of $\beta$ on CoCoOp + Prompt Diffusion.

| $\beta$ | Base | New | H |
|---|---|---|---|
| 0 | 80.47 | 71.69 | 75.83 |
| 0.01 | 81.35 | 74.97 | 78.02 |
| 0.1 | 80.93 | 73.88 | 77.24 |
| 1 | 80.75 | 71.82 | 76.02 |

**Table 15:** Effect of $\beta$ on MaPLe + Prompt Diffusion.

| $\beta$ | Base | New | H |
|---|---|---|---|
| 0 | 82.28 | 75.14 | 78.02 |
| 0.01 | 83.39 | 77.32 | 80.24 |
| 0.1 | 83.03 | 76.81 | 79.80 |
| 1 | 82.74 | 75.97 | 79.21 |

## J EVALUATION UNDER DISTRIBUTIONAL SHIFTS

To address the limitations of existing SOTA prompt methods under different distributional shifts, we conducted a comprehensive evaluation of our method in combination with Xiao et al.'s "Any-Shift Prompting for Generalization over Distributions" (CVPR 2024). We evaluated performance across various types of shifts, including covariate, label, concept, conditional, and multiple shifts. The results are summarized in Tables 16, 17, 18, and 19. From these results, it is evident that our method improves performance across all types of shifts compared to Xiao et al.'s method alone. This comprehensive comparison highlights the limitations of existing SOTA methods in adapting to distributional shifts and underscores the critical importance of our contribution. Specifically, **Prompt Diffusion** enhances instance-level adaptability by refining prompts during inference, thereby addressing the instability and inefficiency caused by fixed prompts under shifting distributions.

**Table 16:** Performance under covariate shifts.

| Method | PACS | VLCS | Office-Home | DomainNet | ImageNet-v2 | ImageNet-S | ImageNet-A | ImageNet-R |
|---|---|---|---|---|---|---|---|---|
| Xiao et al. (2024) | 98.16 | 86.54 | 85.16 | 60.93 | 64.53 | 49.80 | 51.52 | 77.56 |
| + Prompt Diffusion | 99.11 | 87.63 | 86.25 | 62.11 | 65.71 | 51.12 | 52.74 | 78.91 |

**Table 17:** Performance under label shifts.

| Method | Base | New | H |
|---|---|---|---|
| Xiao et al. (2024) | 82.36 | 76.30 | 79.21 |
| + Prompt Diffusion | 83.71 | 78.21 | 80.87 |

**Table 18:** Performance under concept and conditional shifts.

| Method | Concept Shift (ImageNet-superclass) | Conditional Shift (Living-17) | Conditional Shift (Entity-30) |
|---|---|---|---|
| Xiao et al. (2024) | 71.12 | 88.41 | 81.74 |
| + Prompt Diffusion | 73.24 | 90.17 | 83.25 |

**Table 19:** Performance under multiple shifts.

| Method | Art | Clipart | Product | Real | Mean |
|---|---|---|---|---|---|
| Xiao et al. (2024) | 83.40 | 72.53 | 91.24 | 90.84 | 84.50 |
| + Prompt Diffusion | 85.11 | 74.07 | 92.72 | 91.71 | 85.90 |

## K  GENERALIZABILITY TO VIDEO UNDERSTANDING TASKS

To explore the generalizability of our method beyond the image-text domain, we applied our approach to video understanding tasks, specifically using the setup from Ju et al. ("Prompting visual-language models for efficient video understanding," ECCV 2022). We conducted experiments on closed-set action recognition datasets, including HMDB-51, UCF-101, Kinetics-400 (K-400), and Kinetics-700 (K-700). The results, presented in Table 20, are reported in terms of Top-1 accuracy. Our method consistently improves performance across all datasets. This demonstrates that **Prompt Diffusion** can effectively adapt to video tasks, leveraging its ability to generate instance-specific prompts that capture temporal and contextual information unique to video data. However, we recognize that applying our method to other modalities, such as audio or multi-modal tasks, may introduce new challenges. For instance, the nature of sequential and hierarchical dependencies in audio signals may require further adaptations to the diffusion process, such as incorporating domain-specific priors or preconditioning steps for better feature alignment. These experimental results and a discussion of potential challenges and adaptations are included to highlight the versatility of our approach and to address suggestions regarding generalizability beyond image-text tasks.

**Table 20:** Performance on closed-set action recognition datasets.

| Method | HMDB-51 | UCF-101 | K-400 | K-700 |
|---|---|---|---|---|
| Ju et al. (2022) | 66.4 | 93.6 | 76.6 | 64.7 |
| + Prompt Diffusion | **67.3** | **95.1** | **77.8** | **66.3** |

