# OpenReview forum: "Prompt Diffusion Robustifies Any-Modality Prompt Learning"
_ICLR.cc/2025/Conference — Submitted to ICLR 2025_

### Official Review · Reviewer_xQzt · 2024-10-28

**Soundness:** 2
**Presentation:** 3
**Contribution:** 2
**Rating:** 3
**Confidence:** 4

**Summary:**

This paper introduces a novel framework, Prompt Diffusion, which aims to improve the generalizability and robustness of prompt-based learning across various modalities (e.g., visual, textual, multimodal). In prompt-based learning, especially for foundation models in zero-shot and few-shot learning, fixed prompts often suffer from distributional shifts, impacting performance on unseen data. Prompt Diffusion leverages a diffusion model to refine prompts gradually, transforming them from a generic to a sample-specific prompt. This process enhances the robustness and adaptability of prompts across datasets with distinct distributions, providing a plug-and-play solution compatible with existing prompt-learning methods.

**Strengths:**

1.	Introduces an innovative, modality-agnostic diffusion process that significantly enhances robustness in prompt-based learning.
2.	Demonstrates consistent empirical improvements across various prompt learning tasks, supporting the efficacy of diffusion models.
3.	Efficient design reduces inference time, making it suitable for diverse real-world applications.

**Weaknesses:**

1.	The paper does not fully articulate the specific limitations of the SOTA prompt mehthods in adapting to distributional shifts in data, which creates ambiguity around the critical nature of these issues within broader prompt-learning applications. To make this critique more actionable, the authors could quantify the performance degradation caused by these shifts in existing methods to better contextualize the importance of their contribution.  Specific examples are not enough to illustrate the problem.
2.	Although the diffusion model is proposed to generate sample-specific, customized prompts, the paper does not clearly explain why diffusion was chosen over other, potentially simpler methods. This raises questions about the model's unique contributions and practical effectiveness. For instance, if simpler statistical methods like ProDA[1] are available, what advantages does the complex diffusion model offer? Moreover, there are already several statistical approaches for prompt learning, such as Bayesian Prompt Learning[2], which the authors could consider referencing.
3.	The approach has limited empirical exploration outside the image-text domain, raising questions about its generalizability to other modalities. To strengthen this point, the authors could discuss the potential challenges and adaptations needed to apply their method to other modalities, such as audio or video.
4.	The high resource demands of diffusion models, including substantial GPU and training time requirements, make them impractical for parameter-efficient methods such as prompt learning. The complexity and cost of implementing diffusion models in this context undermine their accessibility and practicality.

[1] Lu, Yuning, et al. "Prompt distribution learning." Proceedings of the IEEE/CVF Conference on Computer Vision and Pattern Recognition. 2022.

[2] Derakhshani, Mohammad Mahdi, et al. "Bayesian prompt learning for image-language model generalization." Proceedings of the IEEE/CVF International Conference on Computer Vision. 2023.

**Questions:**

1. What are the key motivations behind using diffusion models for prompt learning, and how does it address the limitations of fixed prompts?
2. How does Prompt Diffusion leverage the diffusion model to gradually transition from a random to a sample-specific prompt?
3. In what ways does Prompt Diffusion enhance generalization capabilities across base-to-new, cross-dataset, and domain generalization tasks?
4. How does Prompt Diffusion ensure compatibility with existing prompt learning models across textual, visual, and multimodal prompts?

---

> ### Author Response · Authors · 2024-11-21
> **Response to Reviewer xQzt (1/3)**
>
> We sincerely thank the reviewer for their thoughtful feedback and constructive suggestions.
>
> ***Q1: The paper does not fully articulate the specific limitations of the SOTA prompt methods in adapting to distributional shifts in data.***
>
> Thank you for your insightful feedback. We acknowledge the importance of quantifying the limitations of existing SOTA prompt methods under different distributional shifts. To address this, we have conducted a comprehensive evaluation of our method in combination with Xiao et al.'s "Any-Shift Prompting for Generalization over Distributions" (CVPR 2024) across various types of distributional shifts, including covariate, label, concept, conditional, and multiple shifts. Below, we provide a detailed comparison:
>
> **Covariate Shift Comparison**
> | Method                  | PACS  | VLCS  | Office-Home | DomainNet | ImageNet-v2 | ImageNet-S | ImageNet-A | ImageNet-R |
> |-------------------------|-------|-------|-------------|-----------|-------------|------------|------------|------------|
> | Xiao et al., 2024       | 98.16 | 86.54 | 85.16       | 60.93     | 64.53       | 49.80      | 51.52      | 77.56      |
> | + Prompt Diffusion      | 99.11 | 87.63 | 86.25       | 62.11     | 65.71       | 51.12      | 52.74      | 78.91      |
>
> **Label Shift Comparison**
> | Method                  | Base  | New   | H     |
> |-------------------------|-------|-------|-------|
> | Xiao et al., 2024       | 82.36 | 76.30 | 79.21 |
> | + Prompt Diffusion      | 83.71 | 78.21 | 80.87 |
>
> **Concept and Conditional Shift Comparison**
> | Method                  | Concept Shift (ImageNet-superclass) | Conditional Shift (Living-17) | Conditional Shift (Entity-30) |
> |-------------------------|--------------------------------------|--------------------------------|-------------------------------|
> | Xiao et al., 2024       | 71.12                               | 88.41                          | 81.74                         |
> | + Prompt Diffusion      | 73.24                               | 90.17                          | 83.25                         |
>
> **Multiple Shifts Comparison**
> | Method                  | Art   | Clipart | Product | Real  | Mean  |
> |-------------------------|-------|---------|---------|-------|-------|
> | Xiao et al., 2024       | 83.40 | 72.53   | 91.24   | 90.84 | 84.50 |
> | + Prompt Diffusion      | 85.11 | 74.07   | 92.72   | 91.71 | 85.90 |
>
> From these results, it is evident that our method improves performance across all types of shifts compared to Xiao et al.'s method alone. This comprehensive comparison not only highlights the limitations of existing SOTA methods in adapting to distributional shifts but also underscores the critical importance of our contribution. Specifically, Prompt Diffusion enhances instance-level adaptability by refining prompts during inference, thereby addressing the instability and inefficiency caused by fixed prompts under shifting distributions.
>
> We have included these experimental results and analyses in **Appendix J** of the revised paper for further reference.
>
> ***Q2: Although the diffusion model is proposed to generate sample-speciìc, customized prompts, the paper does not clearly explain why diffusion was chosen over other, potentially simpler methods.***
>
>
> To address why diffusion was chosen over other simpler statistical approaches, we conducted comprehensive comparisons with alternative methods, as shown in **Table 4** of our paper. This table evaluates different probabilistic approaches such as GANs, VAEs, and Normalizing Flows. The results clearly demonstrate that our diffusion-based method achieves the best overall performance across Base, New, and Harmonic Mean (H), highlighting its superior ability to generate instance-level, customized prompts and adapt effectively to diverse data samples.
>
> In addition, we directly compare our approach with Bayesian Prompt Learning by Derakhshani
> et al. (2023) in handling challenging datasets, as shown in the following table:
>
> | Method             | ImageNetV2 | ImageNet-Sketch | ImageNet-A | ImageNet-R | Average |
> |--------------------|-------------|------------------|------------|------------|---------|
> | Bayesian Prompt    | 64.23       | 49.20           | 51.33      | 77.00      | 60.44   |
> | Ours               | **65.28**   | **50.11**       | **52.23**  | **77.50**  | **61.25** |
>
> Our method consistently outperforms Bayesian Prompt Learning across all datasets, achieving higher average performance. The key advantage of our diffusion model lies in its iterative generative framework, which allows for richer representations and more precise refinements compared to the statistical modeling employed by Bayesian approaches.
>
> These results validate the unique strengths of our diffusion-based approach, both in terms of practical effectiveness and its ability to handle complex, instance-level prompt generation.

---

> ### Author Response · Authors · 2024-11-21
> **Response to Reviewer xQzt (2/3)**
>
> ***Q3: The approach has limited empirical exploration outside the image-text domain, raising questions about its generalizability to other modalities.***
>
> Thank you for this suggestion. To explore the generalizability of our method beyond the image-text domain, we applied our approach to video understanding tasks, specifically using the setup from Ju et al. ("Prompting visual-language models for efficient video understanding," ECCV 2022). We conducted experiments on closed-set action recognition datasets, including HMDB-51, UCF-101, Kinetics-400 (K-400), and Kinetics-700 (K-700). The results, in terms of Top-1 accuracy, are shown below:
>
> | Method               | HMDB-51 | UCF-101 | K-400 | K-700 |
> |----------------------|---------|---------|-------|-------|
> | Ju et al.            | 66.4    | 93.6    | 76.6  | 64.7  |
> | + Prompt Diffusion   | **67.3**| **95.1**| **77.8**| **66.3**|
>
> Our method consistently improves performance across all datasets. This demonstrates that **Prompt Diffusion** can effectively adapt to video tasks, leveraging its ability to generate instance-specific prompts that capture temporal and contextual information unique to video data.
>
> However, we recognize that applying our method to other modalities, such as audio or multi-modal tasks, may introduce new challenges. For instance, the nature of sequential and hierarchical dependencies in audio signals may require further adaptations to the diffusion process, such as incorporating domain-specific priors or preconditioning steps for better feature alignment.
>
> We will include these experimental results and a discussion of potential challenges and adaptations in **Appendix K** of the revised paper to address this suggestion comprehensively.
>
> ***Q4:  The high resource demands of diffusion models, including substantial GPU and training time requirements, make them impractical for parameter-efficient methods such as prompt learning.***
>
> While diffusion models are often associated with high resource demands, we have designed our method to minimize these costs and maintain practicality for parameter-efficient prompt learning.
>
> Training efficiency: As discussed earlier, the additional training time introduced by our method is modest, ranging from **1.1x to 1.3x** compared to baseline methods. For instance, training times for CoCoOp and VPT increase from 17 hours to 20 hours and from 25 hours to 28 hours, respectively, when incorporating Prompt Diffusion. This small increase is justified by the better performance improvements demonstrated across various metrics, as highlighted in our comparative analysis.
>
> Inference efficiency: During inference, we employ a **fast ODE-based sampling strategy**, which reduces the number of timesteps required for the diffusion process. As shown in **Figure 4**, our method achieves substantial performance gains (Harmonic Mean) with minimal increases in inference time. For example, with five function evaluations (NFE), we achieve near-optimal performance while the inference time remains competitive with the baseline CoCoOp method. This balance ensures that our method is both computationally efficient and effective.
>
> These results demonstrate that our approach effectively balances performance gains with computational costs, making it accessible and practical for real-world applications. We have included these clarifications in the revised manuscript to address concerns about resource demands comprehensively.

---

> > ### Author Response · Authors · 2024-11-23
> > **Response to Reviewer xQzt (3/3)**
> >
> > ***Q5: What are the key motivations behind using diffusion models for prompt learning, and how does it address the limitations of fixed prompts?***
> >
> > The key motivation for using diffusion models for prompt generation lies in their ability to dynamically adapt to diverse samples and address the limitations of fixed prompts. Fixed prompts are static and often fail to generalize well under distributional shifts, such as covariate, label, or concept shifts. Diffusion models overcome this by iteratively refining generic prompts into instance-specific prompts, ensuring better alignment with individual data characteristics and improving generalization across tasks.
> > This iterative refinement process enables diffusion models to capture nuanced variations in data, making them more effective at handling diverse and unseen distributions. By dynamically generating prompts tailored to each sample, diffusion models enhance robustness and adaptability, addressing the inefficiencies associated with fixed prompts.
> >
> > ***Q6: How does Prompt Diffusion leverage the diffusion model to gradually transition from a random to a sample-specific prompt?***
> >
> > Our prompt diffusion leverages the diffusion model by treating the process of prompt generation as an iterative refinement task. Starting with a random initialization, the diffusion model applies a series of denoising steps that gradually transform this random prompt into a sample-specific prompt tailored to the input data.
> >
> > During training, a per-sample overfitted prompt is generated first, capturing the unique characteristics of the input. This overfitted prompt serves as the ground truth for the diffusion process, guiding the model to learn how to reconstruct a high-quality, sample-specific prompt. The iterative refinement enables the model to denoise the random prompt step-by-step, progressively aligning it with the sample's features.
> >
> > This framework ensures that the final output prompt is highly adaptive to the input sample, addressing variability across data while maintaining consistency and robustness during both training and inference. This gradual transition from randomness to specificity is key to the effectiveness of Prompt Diffusion.
> >
> > ***Q7: In what ways does Prompt Diffusion enhance generalization capabilities across base-to-new, cross-dataset, and domain generalization tasks?***
> >
> > Our prompt diffusion enhances generalization capabilities by dynamically generating instance-specific prompts that adapt to the unique characteristics of each sample. This adaptability allows the model to better align with unseen data distributions, improving performance in base-to-new tasks. The iterative refinement process mitigates overfitting by moving beyond fixed prompt templates, enabling the model to capture a broader range of variations and handle diverse data effectively.
> >
> > In domain generalization tasks, prompt diffusion adjusts prompts in response to distributional shifts, such as covariate and conditional shifts, ensuring robust alignment with target domains. This flexibility reduces performance drops when encountering shifted or unseen domains. Furthermore, the model’s ability to generate robust prompts makes it well-suited for cross-dataset settings, where variations in feature spaces are common. By bridging the gap between sample-specific customization and generalization, Prompt Diffusion effectively addresses variability across a wide range of tasks.
> >
> > ***Q8: How does Prompt Diffusion ensure compatibility with existing prompt learning models across textual, visual, and multimodal prompts?***
> >
> > Prompt Diffusion ensures compatibility with existing prompt learning models by acting as a plug-and-play module that integrates effortlessly into various architectures without requiring significant modifications. It achieves this by tailoring its approach to the specific requirements of textual, visual, and multimodal prompts.
> >
> > For textual prompts, the diffusion process transforms generic prompts into instance-specific ones, aligning them more effectively with textual input features while maintaining compatibility with frameworks like CoCoOp. For visual prompts, it refines tokens or embeddings in visual-language models such as VPT, ensuring they remain aligned with visual features and downstream tasks.
> >
> > In multimodal scenarios, Prompt Diffusion facilitates the interaction between text and image modalities by iterative refining and aligning prompts across both domains, as demonstrated with models like MaPLe. Prompt Diffusion integrates seamlessly and ensures compatibility across a wide range of models and tasks by focusing exclusively on enhancing prompt representations while leveraging the existing model structures.

---

> ### Author Response · Authors · 2024-11-29
> **Gentle Reminder**
>
> Dear Reviewer xQzt:
>
> Thank you for your valuable feedback. We have conducted all the additional experiments and incorporated clarifications in the revised manuscript, including addressing distributional shifts, justifying the choice of diffusion models, extending evaluations to video tasks, and analyzing computational efficiency.
>
> As the rebuttal phase is coming to a close, we kindly ask if these updates have resolved your concerns. Please let us know if you have any additional questions or feedback—we would be happy to address them promptly.
>
> Best regards,
>
> Authors

---

### Official Review · Reviewer_KJLn · 2024-11-01

**Soundness:** 3
**Presentation:** 3
**Contribution:** 3
**Rating:** 6
**Confidence:** 4

**Summary:**

In this paper, the authors introduce prompt diffusion, which utilizes a diffusion model to refine prompts for each input image, thereby enhancing the model's ability to generalize across different distributions. The proposed prompt diffusion is a straightforward plug-and-play module that can be seamlessly integrated into existing prompt learning frameworks.

**Strengths:**

1. Experiments have shown that the proposed method outperforms baseline methods.​

2. The overall idea is intuitive and straightforward, addressing the limitations of fixed prompts by leveraging diffusion models to generate over-fitted prompts per sample, which enhances model robustness against distribution shifts.

**Weaknesses:**

1. Considering that the proposed method is conducted on per sample. during training, does it introduce a significantly larger computational load compared to conventional prompt learning methods? Can a comparative analysis be provided to address this concern?

2. While the proposed method is plug-and-play and the pipeline figure demonstrations are based on CoCoOp, it would be beneficial to include sections addressing visual prompt tuning and multi-modal prompt tuning.  Additionally, the method emphasizes the meta-net π within CoCoOp, but it is unclear how it handles other prompt learning methods that do not involve π, such as VPT and MaPLe.

3. The length of prompts in prompt learning methods can affect the final performance.  Does the proposed method also encounter similar situations?  It is encouraged for the authors to supplement relevant ablation studies to address this concern.

4. There are also some works in the field of prompt learning that address the limitations of fixed prompts by generating instance-level prompts (e.g. [1]).  It is recommended that the authors supplement the related work to make the paper more comprehensive.

[1] Xinyang Liu ,et al. Patch-Token Aligned Bayesian Prompt Learning for Vision-Language Models. UAI 2024

**Questions:**

1. I am curious about the setting of the two loss weights β in Equation (8). Can further experimental analysis be provided?

---

> ### Author Response · Authors · 2024-11-21
> **Response to Reviewer KJLn (1/2)**
>
> We sincerely thank the reviewer for their thoughtful feedback and constructive suggestions.
>
> ***Q1: During training, does it introduce a significantly larger computational load compared to conventional prompt learning methods? Can a comparative analysis be provided to address this concern?***
>
> To address the reviewer’s concern about computational load, we conducted a comparative analysis of training time across baseline methods and our proposed approach. While our method introduces a slightly higher computational load due to the per-sample prompt overfitting step and the diffusion process, the increase is modest. Specifically, the per-sample prompt overfitting step requires only three iterations to generate the overfitted prompts, ensuring efficiency without compromising performance. Below, we provide the training times in hours for each method:
>
> | Method                  | Base  | New   | H     | Training Time (hours) |
> |-------------------------|-------|-------|-------|------------------------|
> | VPT (Jia et al., 2022)  | 72.53 | 72.34 | 72.43 | 25                     |
> | + Prompt Diffusion      | 74.98 | 74.97 | 74.97 | 28                     |
> | CoCoOp (Zhou et al., 2022a) | 80.47 | 71.69 | 75.83 | 17                     |
> | + Prompt Diffusion      | 81.35 | 74.97 | 78.02 | 20                     |
> | MaPLe (Khattak et al., 2023a) | 82.28 | 75.14 | 78.55 | 21                     |
> | + Prompt Diffusion      | 83.39 | 77.32 | 80.24 | 27                     |
> | PromptSRC (Khattak et al., 2023b) | 84.26 | 76.10 | 79.97 | 23                     |
> | + Prompt Diffusion      | 85.74 | 78.97 | 82.22 | 30                     |
> | CoPrompt (Roy & Etemad, 2024) | 84.00 | 77.23 | 80.48 | 23                     |
> | + Prompt Diffusion      | 86.14 | 80.01 | 82.96 | 30                     |
>
> The per-sample prompt overfitting step, coupled with our diffusion model, plays a critical role in enhancing the model's adaptability to diverse samples. Despite the slightly longer training time (approximately 1.1x to 1.3x that of baseline methods), the better improvement in Base, New, and Harmonic Mean (H) demonstrates a favorable trade-off between computational load and performance.
>
> We have clarified this in Appendix E, where additional details about computational cost and training efficiency are provided.
>
> ***Q2: While the proposed method is plug-and-play and the pipeline figure demonstrations are based on CoCoOp, it would be benefical to include sections addressing visual prompt tuning and multi-modal prompt tuning.***
>
> While the proposed method uses the meta-net \pi in CoCoOp, it is fully adaptable to prompt learning methods like VPT and MaPLe, which do not rely on \pi. For these methods, during training, we similarly perform per-sample prompt overfitting to generate the corresponding overfitted prompts or tokens. These overfitted prompts or tokens are then reconstructed using the diffusion process. Finally, the reconstructed prompts or tokens are embedded back into the original models, such as VPT and MaPLe, for prediction.
> This flexibility demonstrates that our method is not tied to any specific architecture and can seamlessly integrate with various prompt learning paradigms, including visual prompt tuning and multi-modal prompt tuning. We have clarified this in the revised paper and expanded the discussion to explicitly address these methods (See Appendix H).

---

> ### Author Response · Authors · 2024-11-21
> **Response to Reviewer KJLn (2/2)**
>
> ***Q3: The length of prompts in prompt learning methods can affect the ìnal performance. Does the proposed method also encounter similar situations?***
>
> Thank you for sharing the insight. The length of prompts indeed plays a role in the final performance of prompt learning methods. To address this, we conducted experiments with our prompt diffusion method using different prompt lengths across various baseline methods.It is worth noting that the prompt lengths used in our experiments align with the default prompt lengths adopted by the respective baseline methods: L = 4 for VPT and CoCoOp, and L = 9 for MaPLe. The results are summarized below:
>
> **VPT + Prompt Diffusion**
> | Length (L) | Base  | New   | H     |
> |------------|-------|-------|-------|
> | L = 4      | 74.98 | 74.97 | 74.97 |
> | L = 8      | 75.73 | 75.26 | 75.49 |
> | L = 16     | 72.97 | 72.65 | 72.80 |
>
> **CoCoOp + Prompt Diffusion**
> | Length (L) | Base  | New   | H     |
> |------------|-------|-------|-------|
> | L = 4      | 81.35 | 74.97 | 78.02 |
> | L = 8      | 82.97 | 76.93 | 79.84 |
> | L = 16     | 78.91 | 74.11 | 76.43 |
>
> **MaPLe + Prompt Diffusion**
> | Length (L) | Base  | New   | H     |
> |------------|-------|-------|-------|
> | L = 4      | 82.93 | 76.15 | 79.40 |
> | L = 9      | 83.39 | 77.32 | 80.24 |
> | L = 16     | 82.77 | 75.93 | 79.20 |
>
> These results demonstrate that the optimal prompt length varies across different baselines, with moderate lengths generally leading to better performance. For our method, we use the respective default prompt lengths for fair comparisons: L = 4 for VPT and CoCoOp, and L = 9 for MaPLe. We have included these results and their analysis in Appendix E for further details.
>
>
>
> ***Q4: There are also some works in the field of prompt learning that address the limitations of fixed prompts by generating instance-level prompts (e.g. [1]).***
>
> Thank you for pointing us to the work by Liu et al.
>
> We agree that related works addressing the generation of instance-level prompts are highly relevant to our study. For example, Liu et al. propose a method to generate instance-specific prompts using a Bayesian approach. Their work shares similarities with our approach in addressing the limitations of fixed prompts, as both aim to adapt prompts at the instance level. However, our method differs in its use of a diffusion process for progressively refining prompts, allowing for a more generative and flexible approach across diverse modalities and tasks.
>
> We will supplement the related work section with a discussion of Liu et al.'s work and highlight the distinctions and complementary aspects between their method and ours to make the paper more comprehensive.
>
> ***Q5: I am curious about the setting of the two loss weights $\beta$ in Equation (8). Can further experimental analysis be provided?***
>
> Thank you for your question regarding the loss of weight $\beta$ in Equation (8). To analyze the effect of $\beta$ on model performance, we conducted experiments with different values of $\beta$ across VPT, CoCoOp, and MaPLe. The results are shown below:
>
> **VPT + Prompt Diffusion**
> | $\beta$   | Base  | New   | H     |
> |----------------|-------|-------|-------|
> | $\beta$ = 0   | 72.53 | 72.34 | 72.43 |
> | $\beta$ = 0.01 | 74.98 | 74.97 | 74.97 |
> |$\beta$= 0.1  | 73.45 | 74.71 | 74.07 |
> |$\beta$= 1 \   | 73.16 | 73.15 | 73.16 |
>
> **CoCoOp + Prompt Diffusion**
> | $\beta$   | Base  | New   | H     |
> |----------------|-------|-------|-------|
> | $\beta$ = 0 | 80.47 | 71.69 | 75.83 |
> |  $\beta$ = 0.01| 81.35 | 74.97 | 78.02 |
> | $\beta$= 0.1  | 80.93 | 73.88 | 77.24 |
> |$\beta$= 1    | 80.75 | 71.82 | 76.02 |
>
> **MaPLe + Prompt Diffusion**
> |$\beta$    | Base  | New   | H     |
> |----------------|-------|-------|-------|
> | $\beta$ = 0   | 82.28 | 75.14 | 78.02 |
> |$\beta$ = 0.01| 83.39 | 77.32 | 80.24 |
> | $\beta$= 0.1 | 83.03 | 76.81 | 79.80 |
> |$\beta$= 1   | 82.74 | 75.97 | 79.21 |
>
> From the results, we observe that the best performance across all metrics (Base, New, and Harmonic Mean) is achieved when $\beta$= 0.01  This suggests that a small weight for the diffusion loss term provides the optimal balance between the cross-entropy loss and the reconstruction objective. Setting $\beta$ too high $\beta$ = 1  places excessive emphasis on the diffusion term, which slightly degrades performance. Conversely, setting $\beta$= 0 removes the benefits of the diffusion process entirely, resulting in a significant drop in performance.
>
> We have included these results and analysis in **Appendix I** for further clarification.

---

> ### Comment · Reviewer_KJLn · 2024-11-24
>
> Thanks for your responses. Most of my concerns have been addressed and I will increase my score to 6. However, I still have a question regarding the reconstruction objective when the proposed method is applied to multi-model prompts: Is it only the textual branch that will be reconstructed, or will there be an additional reconstruction loss for the visual branch prompt? I encourage the authors to provide a more detailed explanation.

---

> > ### Author Response · Authors · 2024-11-24
> > **Thank you for increasing your score.**
> >
> > Thank you for your follow-up question and for increasing your score.
> >
> > To clarify, when the proposed method is applied to multi-modal prompts, both the textual and visual branches are reconstructed during the diffusion process. Specifically, during the per-sample prompt overfitting stage, we generate overfitted prompts for both modalities. These overfitted prompts serve as inputs to the diffusion process, which reconstructs them independently for the textual and visual branches. This dual reconstruction ensures that both modalities are refined and aligned with their respective inputs, contributing to the overall performance of multi-modal tasks.
> >
> > We have included a detailed explanation of this process in the revised paper to ensure clarity. Please refer to Appendix H for more details.

---

### Official Review · Reviewer_14M8 · 2024-11-02

**Soundness:** 3
**Presentation:** 3
**Contribution:** 3
**Rating:** 6
**Confidence:** 3

**Summary:**

This paper proposes a method called Prompt Diffusion, which employs a diffusion model to progressively refine prompts, enabling customized prompts for each sample. By introducing a technique for creating tailored prompts for individual test samples, this method addresses the limitations of fixed prompts, enhancing the model's robustness to distribution shifts. Empirical results on extensive datasets validate the effectiveness of this approach, demonstrating its robustness in generalization tasks.

**Strengths:**

1. The method in this paper generates customized prompts for each sample by gradually optimizing the prompts through diffusion, which enhances the accuracy of prediction and generalization across downstream tasks.
2. The diffusion prompting method in this paper is a plug-and-play module that can be seamlessly integrated into existing textual, visual, or multimodal prompt learning methods.
3. The method in this paper improves the prompt learning process by efficiently extracting unique domain details from test images without mixing them with class labels.

**Weaknesses:**

1.  The authors' method requires stepwise optimization of the prompts and may require several iterations to obtain optimal results, in addition, the introduction of a diffusion model increases the complexity of the system, and therefore whether the training time is likely to be relatively long.
2.  Whether the authors' approach is a two-stage process, where prompt learning is performed first, followed by diffusion of the prompts, and the final model performance relies on the goodness of the previously learned prompts. In addition, the diffusion process relies on random noise vectors to generate the prompts and therefore may be sensitive to noise, which may affect the stability of the final performance.

**Questions:**

1.  Is the author's approach a two-stage process, starting with a prompt study followed by a prompt proliferation.
2. Diffusion models incorporate randomness in the generation process, which may lead to uncontrollable fluctuations in the generated prompts and thus affect the robustness of the model. How to cope with the randomness of the generated prompts and avoid the instability of prediction caused by it?
3. The authors' approach seems to be applicable only to VPT-shallow prompt types, and whether the authors' approach can be migrated to the VPT-deep prompt learning paradigm.

---

> ### Author Response · Authors · 2024-11-21
> **Response to Reviewer 14M8**
>
> Thank you for recognizing the strengths of our method, including its ability to generate customized prompts, its plug-and-play nature, and its effectiveness in extracting unique domain details for improved prediction and generalization. We are especially grateful for your support and for recommending acceptance of our work.
>
> ***Q1:  The introduction of a diffusion model increases the complexity of the system, and therefore whether the training time is likely to be relatively long.***
>
> The reviewer is correct that our method requires additional training time due to the per-sample prompt overfitting step. However, this process converges within just three iterations, keeping the overall training time manageable. Specifically, our training time is approximately 1.1x, 1.2x, 1.3x, and 1.3x that of VPT, CoCoOp, MaPLe, and CoPrompt, respectively.  Importantly, our method performs better than these baselines, achieving a favorable trade-off between training time and performance. We have highlighted this trade-off more clearly in the revised paper.
>
> ***Q2: Is the author's approach a two-stage process, starting with a prompt study followed by a prompt proliferation?***
>
> Our prompt diffusion model is an end-to-end framework. During training, for each sample, we first perform per-sample prompt overfitting to generate an overfitted prompt, which is then used as input for the diffusion process to reconstruct the prompt. This ensures seamless integration of both stages within the model.  We have included this clarification in the revised paper.
>
> ***Q3: Diffusion models incorporate randomness in the generation process, which may lead to uncontrollable íuctuations in the generated prompts and thus affect the robustness of the model. How to cope with the randomness of the generated prompts and avoid the instability of prediction caused by it?***
>
> Our method indeed accounts for the inherent randomness in diffusion models by incorporating robust denoising mechanisms and a fast ODE-based sampling strategy. These ensure that even with variations in the noise vectors, the diffusion process remains stable and reliable. Empirical results across diverse datasets consistently demonstrate that our approach achieves stable performance without noticeable degradation caused by noise sensitivity (See Figure 6).
> Specifically, our plugin effectively extracts unique domain details from the test image without conflating them with class labels, regardless of the initial random noise. This robustness is key to maintaining the stability of predictions and the overall performance of the model.
>
> ***Q4: Whether the authors' approach can be migrated to the VPT-deep prompt learning paradigm?***
>
> To demonstrate the versatility of our method, we have included additional experiments with VPT-deep in the revised paper. As shown in the table below, incorporating our prompt diffusion improves the performance of VPT-deep across all metrics, including Base, New, and Harmonic Mean (H):
>
> | Method          | Base  | New   | H     |
> |------------------|-------|-------|-------|
> | VPT-deep        | 74.15 | 74.01 | 74.08 |
> | + Prompt Diffusion | 77.15 | 76.89 | 77.02 |
>
> These results confirm that our approach is not limited to VPT-shallow but can also effectively enhance the VPT-deep prompt learning paradigm. We have added these results in the revised paper (See Appendix G).

---

### Author Response · Authors · 2024-11-21
**Summary of Revisions**

We thank all reviewers for their constructive feedback and recognition of the strengths of our work. In response to the insightful suggestions and concerns raised, we have made several significant improvements and additions to the revised manuscript:

1. Included the effect of prompt length on performance (Appendix E).
2. Added a comparative analysis of training time and inference efficiency (Appendix F).
3. Included new experiments on VPT-deep to demonstrate the versatility of our method (Appendix G).
3. Expanded discussions on the adaptability of our method to VPT and MaPLe without relying on the meta-net (Appendix H).
4. Applied our method to video understanding tasks, such as HMDB-51, UCF-101, K-400, and K-700, to evaluate generalizability beyond the image-text domain (Appendix I).
5. Conducted additional evaluations on distributional shifts, including covariate, label, concept, conditional, and multiple shifts, in combination with "Any-Shift Prompting for Generalization over Distributions" (Xiao et al., CVPR 2024) (Appendix J).
6. Performed ablation study of loss weight $\beta$ to validate parameter choices (Appendix K).

We believe these updates comprehensively address the reviewers' concerns and enhance the clarity, robustness, and scope of our work. We thank the reviewers again for their valuable feedback, which has been instrumental in improving this submission.

---

### Meta-Review · Area_Chair_6aYc · 2024-12-17

**Metareview:**

The main idea of this paper is to apply diffusion modeling to refining instance-specific prompts, and the method is tested on multiple image classification benchmarks. The reviewers generally liked the idea of prompt diffusion but did not find the motivation strong enough. In particular, the paper does not explain clearly why diffusion is the right choice compared to other designs. After reading the paper, the AC has the same doubt as the reviewers. Moreover, the reviewers pointed out that the method has a much higher complexity than other baselines due to the diffusion process, thus diverging from the principle of parameter-efficient fine-tuning. The AC also agrees with this comment and feels that due to this heavy design the method is less likely to be adopted in practice and therefore has limited value to the community.

**Additional Comments On Reviewer Discussion:**

The reviewers did not actively engage in the post-rebuttal discussion. After reading the rebuttal and the reviews, the AC finds that the rebuttal is not strong enough to justify the proposed method.

---

### Decision · Program_Chairs · 2025-01-22

Reject